# Adrenomedullin restores the human cortical interneurons migration defects induced by hypoxia

Alyssa Puno[1†], Wojciech P Michno[1†], Li Li[1], Amanda Everitt[2], Kate McCluskey[2], Saw Htun[1], Dhriti Nagar[1], Jong Bin Choi[1], Yuqin Dai[3], Seyeon Park[1], Emily Gurwitz[1], Jeremy A Willsey[2,4], Fikri Birey[5], Anca M Pasca[1*]

[1]Department of Pediatrics, Stanford University, Stanford, United States; [2]Department of Psychiatry and Behavioral Sciences, UCSF Weill Institute for Neurosciences, University of California, San Francisco, San Francisco, United States; [3]Sarafan ChEM-H, Stanford University, Stanford, United States; [4]Quantitative Biosciences Institute (QBI), University of California, San Francisco, San Francisco, United States; [5]Department of Human Genetics, Emory University, Atlanta, United States

## eLife Assessment

The authors combined human assembloids, fetal brain tissue, bulk and single cell RNA sequencing, and live imaging to understand the molecular mechanisms affected by hypoxia during cortical development. The findings are very **important** to the neurodevelopmental field, They reveal new insights into how migration of cortical interneurons can be affected in hypoxic conditions, and provide exciting models to probe broad neurodevelopmental processes in health and disease. The evidence is **compelling**. The data and analyses are very rigorous and go beyond the state-of-the-art.

*For correspondence:
apasca@stanford.edu

†These authors contributed equally to this work

**Abstract** Extremely preterm birth (at <28 postconceptional weeks) leads to brain injury and represents the leading cause of childhood-onset neuropsychiatric diseases. No effective therapeutics exist to reduce the incidence and severity of brain injury of prematurity. Hypoxic events are the most important environmental factor, along with inflammation. Among other developmental processes, the second half of in utero fetal development coincides with the migration of cortical interneurons from the ganglionic eminences into the cortex; this process is thus prone to disruptions following extremely preterm birth. To date, no studies have directly investigated the migration of human cortical inhibitory neurons under hypoxic conditions. Using multi-day confocal live imaging in human forebrain assembloids (hFA) derived from human-induced pluripotent stem cells (hiPSCs) and ex vivo developing human brain tissue, we found a substantial reduction in the migration of hypoxic interneurons. Using transcriptomics, we identified adrenomedullin (*ADM*) as the gene with the highest fold change increase in expression. Based on previous literature about the protective role of supplemental ADM for other injuries, here, we demonstrated that addition of exogenous ADM to the hypoxic media restores the migration defects of interneurons. Lastly, we showed that one of the mechanisms of protection by ADM is through the activation of the cAMP/PKA pathway and subsequent pCREB-dependent rescued expression of a subset of GABA receptors, which are known to promote migration. Overall, in this manuscript, we provide the first direct evidence for hypoxia-induced deficits in the migration of human cortical interneurons and identify ADM as a possible target for therapeutic development.

**eLife digest** Medical advances have significantly improved the survival of preterm and extremely preterm babies (less than 28 weeks of pregnancy). However, no therapies yet exist to prevent brain injury that is often associated with preterm birth, often leading to lifelong neurological and psychiatric disorders.

Over the past few decades, research studies have used rodent models to try to identify therapeutics. But when it came to testing in humans, therapies that looked promising in rodents failed to show benefit in patients, suggesting possible interspecies differences and the need to study human cells. While this has been challenging in the past, recent major scientific advances now allow the generation of human brain cells from stem cells, which have themselves been derived from human skin or blood cells.

Clinical studies have shown that hypoxia – oxygen deprivation of the brain – is a major risk factor for brain injury of prematurity. Brain tissue studies of people previously born preterm have shown a decrease in the number of grey matter inhibitory neurons, which help prevent overactivity in the healthy brain, such as seizures. During fetal development, these cells migrate long distances to reach their final location for optimal activity. Puno et al. tested whether hypoxia affects the migration of inhibitory neurons and if any damage could be reversed. They used human brain assembloids, which are three-dimensional clumps of brain cells that recreate the cell-migration process in a laboratory dish.

Puno et al. found that hypoxia significantly decreases the migration of inhibitory neurons and that this injury persists even after re-adding oxygen to the cells. Using advanced research tools, they found that a protein called adrenomedullin increases in the organoids in response to hypoxia, but due to a lack of oxygen it cannot be modified to make it functional. However, adding more functional adrenomedullin to the cells during hypoxia restores migration.

Puno et al. show for the first time that migration of inhibitory neurons is affected by hypoxia. They identify adrenomedullin as a potential target for developing effective therapies for premature babies, which may also be relevant for adults with hypoxic brain injuries such as stroke. Further studies in animal models that are more similar to humans are needed to confirm adrenomedullin's safety, dosage and mechanism of action.

## Introduction

Extremely preterm birth (at <28 postconceptional weeks) is commonly associated with brain injury of prematurity and represents the leading risk factor for childhood-onset neuropsychiatric diseases (*Moster et al., 2008*; *Volpe, 2009a*; *Crump et al., 2021*). Clinical studies have identified postnatal hypoxic events as one of the most important environmental risk factors for brain injury of prematurity (*Volpe, 2009b*). Despite important clinical progress in understanding brain injury of prematurity from a histologic and radiologic perspective, the cell-type-specific biological mechanisms of injury remain largely understudied in humans.

Among other phenotypes (*Penn et al., 2016*), histology studies demonstrate a substantial reduction in the number of cortical interneurons in post-mortem tissue samples from individuals previously born extremely premature (*Lacaille et al., 2019*). This decrease is now suggested as a significant contributor to the pathophysiology of neuropsychiatric disorders associated with preterm birth (*Salmaso et al., 2014*; *Iai and Takashima, 1999*; *Robinson et al., 2006*; *Takano, 2015*). Cortical interneurons are born and proliferate in the ganglionic eminences before migrating to the cortex, where they eventually become part of the cortical synaptic networks in a process that evolves over multiple years in humans (*Paredes et al., 2016*). Of all these developmental stages, migration toward the cortex is the process most active during the second half of in utero development, which, in extremely preterm infants, is replaced by postnatal hospitalization in the Neonatal ICU and associated with hypoxic events due to lung immaturity and inability to transfer oxygen from the environment to the tissues.

Despite migration defects being highly suspected to be impaired in hypoxic brain injury of prematurity, the investigation of this process in preclinical models has been uniquely challenging to study, because this migration mostly happens in utero in most species. As a result, to date, no studies have yet been able to directly visualize the migration patterns of cortical interneurons upon exposure

to hypoxia. Moreover, no in-depth data exist about the molecular mechanisms of injury in hypoxic migrating interneurons, thus making therapeutic discovery challenging. Previous animal models for hypoxic brain injury of prematurity have, however, replicated the human-observed histologic decrease in cortical interneurons in the dorsal forebrain of pre-born embryos as early as 4 days after exposing the pregnant dam to a hypoxic injury and suggested that this decrease is related to a defect in the migration of interneurons that have already exited the ganglionic eminences at the time of hypoxia exposure (*Nisimov et al., 2018*). These findings are encouraging and support the hypothesis that migration is affected by hypoxia and highlight the need for further investigation using newly emerging scientific tools.

To study the effects of hypoxia on the migration of cortical interneurons and advance our mechanistic understanding of prematurity-associated interneuronopathies, we developed two complementary human cellular models that consist of a stem-cell-based in vitro human cellular platform and an ex vivo developing human brain tissue platform. Specifically, we established a multi-day confocal live imaging setup for the visualization of migrating cortical interneurons within human forebrain assembloids (hFAs) derived from induced pluripotent stem cells (hiPSCs) (*Sloan et al., 2018*; *Birey et al., 2017*), and within ex vivo developing human brain tissue.

Using these new models, we identified substantial migration deficits during exposure to hypoxia, thus providing the first direct evidence for the long-proposed migration delays induced by hypoxia. These findings are important because delays in reaching the specified cortical targets within the developmentally appropriate window are known to lead to elimination of cortical interneurons through apoptosis, and the direct demonstration of this deficit strengthens the hypothesis that delayed migration contributes to the decrease in the number of cortical interneurons observed in post-mortem brain tissues from preterm individuals.

Additionally, we demonstrated that supplementation with exogenous ADM peptide restores the migration of cortical interneurons during exposure to hypoxia. Adrenomedullin (ADM) peptide has been previously shown to acutely increase in various non-neurologic and neurologic hypoxic and inflammatory conditions, in both clinical and preclinical studies (*Ishiyama et al., 2023*; *Valenzuela-Sánchez et al., 2016*; *Solé-Ribalta et al., 2022*). Moreover, it has been demonstrated to be a direct binding partner for HIF1α, which is a hallmark of hypoxia, but is also stabilized in inflammation (*Nguyen and Claycomb, 1999*). Interestingly, ongoing clinical trials for the treatment of inflammatory bowel disease have shown therapeutic promise and safety upon administration of exogenous ADM at supraphysiological levels (*Kita et al., 2022*; *Zhang et al., 2009*; *Pedreño et al., 2014*; *Gao et al., 2018*). Our data, in conjunction with previous findings, suggest exogenous administration of ADM as a therapeutic intervention should be further investigated in vivo and for various hypoxic injuries.

## Results

### Development of a human cellular model to study the migration of hypoxic cortical interneurons

The hFAs recapitulate key developmental aspects of cortical interneurons migration from the medial ganglionic eminences (MGE) into the dorsal forebrain. To study the effects of hypoxia on the migration of human cortical interneurons, we established a model for hypoxic injury using hFAs. Using standardized published protocols, we previously contributed to developing (*Sloan et al., 2018*; *Birey et al., 2017*), we generated hFAs through the differentiation of hiPSCs into human cortical organoids (hCO) containing dorsal forebrain excitatory neurons (*Paşca et al., 2015*), and human subpallial organoids (hSO) containing cortical interneurons (*Birey et al., 2017*). To visualize migrating cortical interneurons, we first fluorescently tagged cells within hSOs with a forebrain interneurons cell-type-specific lentiviral reporter (Dlxi1/2b::eGFP; *Potter et al., 2009*). Subsequently, we fused the labeled hSOs with hCOs to generate hFAs. At approximately 10–14 days after fusion into hFAs, we observed substantial and active migration of Dlxi1/2b::eGFP⁺ cortical interneurons on the hCO side of hFAs (*Figure 1A*, *Figure 1—figure supplement 1A*).

To monitor the migration patterns of cortical interneurons under control and hypoxic stress, we established a long-term live imaging microscopy setup in intact hFAs using a confocal microscope with a motorized stage (Zeiss LSM 980 with Airyscan 2) equipped with an environmentally controlled chamber (37 °C, 5% $CO_2$) and an oxygen level controller (Okolab Bold Line). For this, the hFAs were

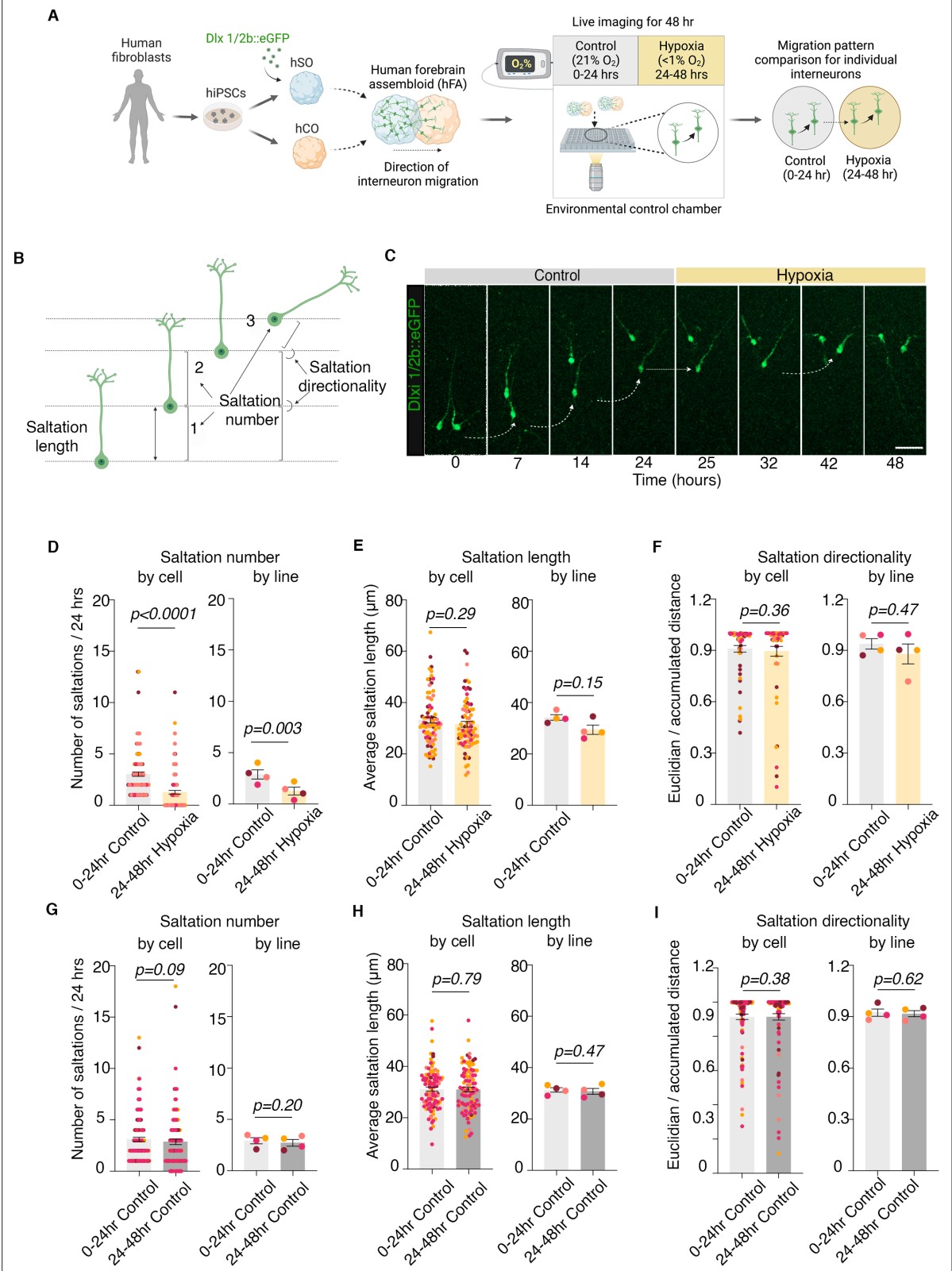

**Figure 1.** Human cellular model to study migration patterns in cortical interneurons under hypoxic stress. (**A**) Schematic illustrating overall experimental design: hiPSCs were used to derive human cortical organoids (hCO) and human subpallial organoids (hSO); for direct visualization of migrating interneurons, at ~45–55 days in culture hSO were infected with lentivirus Dlxi1/2b::eGFP and then fused with hCO into human forebrain assembloid (hFA); to study interneuron migration during exposure to hypoxia, hFA were imaged 10–14 days post infection using confocal live-imaging setup

*Figure 1 continued*

focused on the hCO part of the hFA, the movement of the same cells was followed for a total of 48 hr, each condition for 24 hr: 0–24 hr in control and 24–48 hr in hypoxia; Created with BioRender.com.(**B**) Schematic of migratory pattern of interneurons, focused on number of saltations, average saltation length, and directionality; (**C**) Example of migration pattern of one cortical interneuron during control and hypoxic conditions; Created with BioRender. com. (**D**) Quantification of saltations number/24 hr in hypoxia-exposed and non-exposed cortical interneurons by individual cells (paired Wilcoxon test, p<0.0001) and by hiPSC line (two-tailed paired *t*-test, p=0.003); (**E**) Quantification of average saltation length for hypoxia-exposed and non-exposed cortical interneurons by individual cells (paired Wilcoxon test, p=0.29) and by hiPSC line (two-tailed paired *t*-test, p=0.15); (**F**) Quantification of directionality of migration in hypoxia-exposed and non-exposed cortical interneurons by individual cells (paired Wilcoxon test, p=0.36), and by hiPSC cell line (two-tailed paired *t*-test, p=0.47); (**G**) Quantification of saltations number/24 hr in control conditions, in the first 24 hr (0–24 hr) versus the subsequent 24 hr (24–48 hr) of live imaging by individual cells (paired Wilcoxon test, p=0.09) and hiPSC line (two-tailed paired *t*-test, p=0.2); (**H**) Quantification of average saltation length in control conditions, in the first 24 hr (0–24 hr) versus the subsequent 24 hr (24–48 hr) of live imaging by individual cells (two-tailed paired *t*-test, p=0.79) and by hiPSC line (two-tailed paired *t*-test, p=0.47); (**I**) Quantification of directionality of migration in control conditions, in the first 24 hr (0–24 hr) versus the subsequent 24 hr (24–48 hr) of live imaging by individual cells (paired Wilcoxon test, p=0.38) and by hiPSC line (two-tailed paired *t*-test, p=0.62); Bar charts: mean ± s.e.m; scale bar: 50 μm.

The online version of this article includes the following source data and figure supplement(s) for figure 1:

**Figure supplement 1.** Example of hFA, oxygen level measurements, qPCR changes in expression of hypoxia-responsive genes and cell death analyses.

**Figure supplement 1—source data 1.** Original files for western blot analysis are shown in *Figure 1C*.

**Figure supplement 1—source data 2.** PDF file containing original western blots *Figure 1C*, including the relevant bands, iPSC lines, and experimental conditions.

transferred to the confocal microscope in a glass-bottom 96-well plate with fresh cell culture media. We imaged individual Dlxi1/2b::eGFP-tagged interneurons within the hCOs for a total of 48 hr: 24 hr (0–24 hr) under control conditions (37 °C, 5% $CO_2$, 21% $O_2$), followed by 24 hr (24–48 hr) in pre-equilibrated hypoxic media (37 °C, 5% $CO_2$, 94.5% $N_2$,<1% $O_2$; *Figure 1A*).

To validate the decrease in partial pressure of $O_2$ ($PO_2$) in the culture media in hypoxic conditions, we used an oxygen optical microsensor OXB50 (50 μm, PyroScience) attached to a fiberoptic multi-analyte meter (FireStingO₂, PyroSciences). In line with our previous reports (*Pașca et al., 2019*), we observed a decrease in the media $PO_2$ level from ~150 mmHg in control conditions to ~25–30 mmHg in the confocal microscope environmental chamber in hypoxic conditions (*Figure 1—figure supplement 1B*).

Next, we confirmed that these conditions are sufficient to activate the hypoxia response pathway in hSOs within 24 hr of exposure, without inducing extensive death of cortical interneurons. Specifically, to validate the induction of hypoxia, we demonstrated the stabilization of HIF1α hypoxia-marker protein using western blot analyses in hSOs derived from four individual cell lines (two-tailed paired *t*-test, p=0.01; *Figure 1—figure supplement 1C, D*). Separately, we validated the activation of the hypoxia-response pathway using targeted transcriptional analyses for hypoxia-responsive genes, including *PFKP* (two-tailed paired *t*-test, p=0.004), *PDK1* (two-tailed paired *t*-test, p=0.001), *VEGFA* (two-tailed paired *t*-test, p=0.003; *Figure 1—figure supplement 1E*, *Supplementary file 2*). To check whether Dlxi1/2b::eGFP-tagged interneurons start expressing cell markers for cell death, we used two different assays: we performed FACS-based Annexin V analyses in fluorescently tagged interneurons, as well as immunostainings for cleaved caspase 3 (c-CAS3). We found no significant difference between control and hypoxic conditions using Annexin V (unpaired *t*-test, p=0.362; *Figure 1—figure supplement 1F*) nor c-CAS3 (unpaired *t*-test, p=0.125) (*Figure 1—figure supplement 1G, H*).

These results demonstrate that our target hypoxic conditions ($PO_2$ ~25–30 mmHg) are sufficient to induce and activate the hypoxia response pathway, but do not induce cell death. This experimental design aligns with our goal of targeting mild hypoxic stress, partly because mild hypoxic episodes are much more common than the severe ones in preterm infants, and partly because we wanted to define, for the first time, the altered molecular pathways in hypoxic (but live) cortical interneurons.

## Human cortical interneurons display migration deficits upon exposure to hypoxia

The migration of cortical interneurons from the ganglionic eminences into the dorsal forebrain is at its peak during the second half of pregnancy in humans (*Anderson et al., 1997*). It is characterized by repetitive cycles of nuclear saltations (nucleokinesis events) in the direction of the leading process (*Mayer et al., 2018*; *Lim et al., 2018*). To study migration patterns of hypoxic cortical interneurons,

we followed the movement of the same individual Dlxi1/2b::eGFP⁺ cortical interneurons for a total of 48 hr (control conditions: hr 0–24; hypoxia: hr 24–48). We analyzed the number of nuclear saltations, saltation length, and the directionality of the migration in interneurons that had already migrated into the hCO at the time of the hypoxia exposure (*Figure 1B and C*).

To quantify the number of saltations, we focused on interneurons that had at least one saltation within the first 24 hr under control conditions and remained visible throughout the entire 48 hr of imaging. This experimental approach was specifically chosen to focus the analyses on actively migrating cells, while excluding those that may have ceased migration and begun integrating into synaptic networks at the time of imaging. To quantify saltation length and directionality, we narrowed the cell population to the subset of cells that had at least one saltation in both control and hypoxic conditions, as a minimum of 1 saltation/condition is a prerequisite for direct comparisons for these parameters.

To identify defects in the migration patterns of interneurons in hypoxia, we analyzed migrating interneurons for 24 hr in control conditions (hr 0–24), followed by another 24 hr in hypoxic conditions (hr 24–48). We observed a ~58% decrease in the number of saltations of individual interneurons (paired Wilcoxon test, p<0.0001), and demonstrated that the phenotype is present in hFAs derived using hiPSCs from four separate individuals (two-tailed paired *t*-test, p=0.003; *Figure 1D*). Importantly, 70 out of 129 actively migrating cortical interneurons not only decreased the number of saltations but became completely stationary upon exposure to hypoxia for 24 hr. However, we found no significant differences in the average length of saltation of cells when analyzed by individual cells (paired Wilcoxon test, p=0.29) and by hiPSC line (two-tailed paired *t*-test, p=0.15; *Figure 1E*). In addition, we did not observe any differences in the directionality of migration measured by the ratio of Euclidean/accumulated distance when analyzed by individual cells (paired Wilcoxon test, p=0.36) and by hiPSC line (two-tailed paired *t*-test, p=0.47; *Figure 1F*).

To ensure that the confocal chamber environment during the prolonged live imaging does not result by itself in defects in migration of the cortical interneurons, or that a possible hypoxia phenotype could be related to a physiologic saltatory pause in the second half of imaging, we performed two separate types of experiments. First, we compared the migration patterns in control conditions (37 °C, 5% $CO_2$, 21% $O_2$) in the first 24 hr (hr 0–24) versus the second 24 hr (hr 24–48). We identified no differences in the total number of saltations/24 hr when analyzed by individual cells (paired Wilcoxon test, p=0.09) and hiPSC line (two-tailed paired *t*-test, p=0.2; *Figure 1G*); no difference in average saltation length when analyzed by individual cells (two-tailed paired *t*-test, p=0.79) and by hiPSC line (two-tailed paired *t*-test, p=0.47; *Figure 1H*); no difference in directionality of migration when analyzed by individual cells (paired Wilcoxon test, p=0.38) and by hiPSC line (two-tailed paired *t*-test, p=0.62; *Figure 1I*). Second, to ensure that the deliberate exclusion from analyses of cortical interneurons that display no saltations in control conditions does not bias the migration phenotype by excluding cells that might otherwise migrate more in the 24–48 hr of imaging, we identified all cortical interneurons that displayed no migration in control conditions (0–24 hr) and quantified their migration in hypoxia. Out of 113 total cells, 108 (95.6%) remained stationary (*Figure 1—figure supplement 1I*), suggesting these cells are not in a highly migratory phase at the time of imaging.

Overall, the experiments in *Figure 1*, *Figure 1—figure supplement 1* suggest mild hypoxia preferentially affects the total migration length of cortical interneurons by decreasing the number of saltations; however, it does not interfere with the length of individual saltations nor the directionality of the migration. The additional validation experiments demonstrate the robustness of the experimental design and strengthen the validity of saltation deficit phenotype in hypoxia. Details on the number of cells, experiments, and number of hiPSC lines used for each experiment are presented in *Supplementary file 1*.

## Transcriptional analyses in hypoxic hSOs suggest ADM as a key component of hypoxia-induced cellular response

To investigate the transcriptional changes induced by mild hypoxia in hSOs, we performed bulk transcriptome-wide RNA-Sequencing (RNA-Seq) at 46 days in culture (before fusion with hCOs), following exposure to hypoxia for 12 hr or 24 hr, as well as at 72 hr after reoxygenation (*Figure 2A*). For these experiments, we induced hypoxia using the C-chamber hypoxia sub-chamber (Biospherix). We confirmed a significant decrease in $PO_2$ between control and hypoxia conditions (two-tailed paired

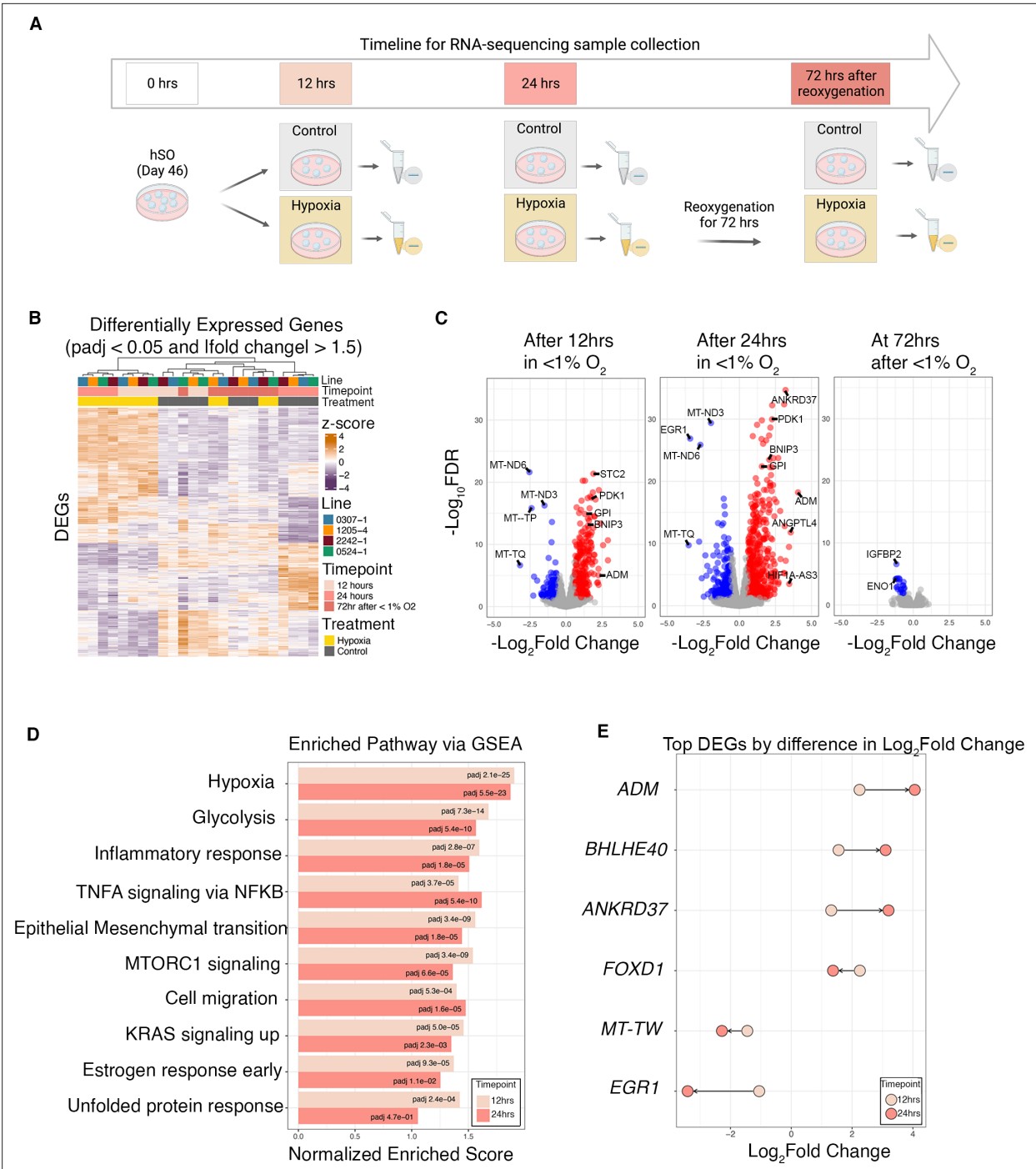

**Figure 2.** Transcriptional changes in hSO exposed to hypoxia. (**A**) Schematic of hypoxia exposure of hSO and collection of samples for RNA-Sequencing; Created with BioRender.com. (**B**) Heatmap of differentially expressed genes in RNA-Seq data showing clear transcriptional changes in hypoxia-exposed samples. Samples (n = 24) from hSO differentiated from four hiPSC lines were collected at 12 hr and 24 hr of exposure to hypoxia, as well as 72 hr after reoxygenation. The union of all differentially expressed genes (n=1473) and samples (n=24) is ordered by hierarchical clustering (complete linkage of Euclidean distance). Z-score normalized expression values are depicted on a continuous scale from lower values (purple) to higher values (orange). Cell lines, treatment, and time point are depicted at the top and represented by different colors; (**C**) Volcano plots of differentially expressed genes (DEGs) at 12 hr, 24 hr, and 72 hr after reoxygenation. Each dot represents a single gene. DEGs with a padj <0.05 and an absolute fold change >1.5 are shown in red (upregulated) or blue (downregulated) and unchanged genes are shown in gray; (**D**) Bar plot of the top 10 shared enriched gene pathways across hypoxic conditions. Adjusted p-values are depicted as text and colors represent the length of hypoxic exposure; (**E**) Dumbbell plot of the top 6 DEGs with the largest positive difference and the largest negative difference in Log$_2$ fold change between 12 and 24 hr of hypoxia exposure; the arrow indicates the direction of change from 12 to 24 hr. (**F**) Transcriptional upregulation (by qPCR) of *ADM* gene in hSO samples exposed

*Figure 2 continued on next page*

Figure 2 continued

to 24 hr of hypoxia: *ADM* (two-tailed paired *t*-test, p=0.02); (**G**) Quantitative enzyme immunoassay analysis of adrenomedullin (ADM) peptide in media from hSO exposed and non-exposed to hypoxia (unpaired Mann-Whitney test, p<0.0001); for values below the minimal detection range of <0.01 ng/mL, value was approximated to 0 (we had 9 values approximated to 0 in the control samples). Bar charts: mean ± s.e.m.; Different dot colors represent individual hiPSC lines.

The online version of this article includes the following figure supplement(s) for figure 2:

**Figure supplement 1.** Oxygen level measurements for RNA-Sequencing experiments and dendrogram of sample clustering.

*t*-test, p<0.0001; *Figure 2—figure supplement 1A*), and showed that $PO_2$ levels were similar to the ones measured in the confocal environmental chamber (*Figure 1B*).

Hierarchical clustering of the gene expression profiles showed clear separation between hSOs from the control (control 12 hr, control 24 hr, control 72 hr), hypoxia-exposed (hypoxia 12 hr, hypoxia 24 hr), and hypoxia followed by reoxygenation samples (hypoxia 72 hr after <1%; *Figure 2—figure supplement 1B*), suggesting that hypoxia exposure induces strong transcriptional profile changes in hSOs, and these largely recover following 72 hr of reoxygenation. For this experiment, we used hSOs derived from 4 individual hiPSC lines, two XX and two XY genotypes (details by hiPSC line are presented in *Supplementary file 1*).

Next, we identified differentially expressed genes (DEGs) between control, hypoxia-exposed, and reoxygenated samples (padj <0.05, fold change >1.5), controlling for potential confounding variables as described in detail in the methods section. We identified 734 differentially expressed genes at 12 hr, 985 genes at 24 hr, and 21 genes at 72 hr after reoxygenation (*Figure 2B and C*, *Supplementary file 3*).

Gene set enrichment analyses (GSEA) demonstrated changes in molecular pathways associated with exposure to hypoxia (including glycolysis, mTORC1, KRAS), inflammatory response pathways (represented by nonspecific immune-related cell surface receptors), estrogen response pathways, and unfolded protein response. In addition, we identified changes in genes associated with general cellular migration (e.g. *AXL*, *GLIPR1*, etc; *Figure 2D*; *Supplementary file 2*).

Interestingly, we found that the *ADM* gene had the highest fold change among DEGs at 24 hr of hypoxia exposure ($log_2FC = 4.06$, padj = 6.08E-19; *Figure 2E*). To validate the increase in gene expression level of *ADM* under hypoxic conditions observed in bulk RNA-sequencing, we performed qPCR analyses using hypoxia-exposed intact hSOs. We confirmed the significant transcriptional upregulation of *ADM* gene expression (two-tailed paired *t*-test, p=0.02; *Figure 2F*).

To check whether the transcriptional upregulation of *ADM* gene expression translates into changes at a protein level, we quantified ADM peptide levels using quantitative enzyme immunoassay (EIA; Phoenix Pharmaceuticals, EK-010–01) in cell culture media from intact hSOs exposed and non-exposed to hypoxia for 24 hr. We observed a significant increase in the protein concentration in the hypoxia-exposed hSOs compared to control conditions (unpaired Mann-Whitney test, p<0.0001; *Figure 2G*). This experiment was performed in three separate biological replicates of hSOs, each derived from four hiPSC lines in one differentiation experiment (details by hiPSC line are presented in *Supplementary file 1*).

## ScRNA-sequencing in hypoxic hFAs uncovers cell type-specific expression of *ADM* and its receptors

To bring novel insight into the cell-type-specific expression of *ADM* under control and hypoxic conditions in human brain cells, we performed single-cell RNA-sequencing in hSOs and hCOs, separated from hFAs that were previously exposed or non-exposed to hypoxia. To do this, we cut the hFAs under microscopic guidance (*Figure 3A*). We used hFAs derived from two different hiPSC lines.

First, we performed Uniform Manifold Approximation and Projection (UMAP) dimensionality reduction for hSOs non-exposed (n=8691 cells) and exposed to hypoxia (n=7331 cells). We identified five distinct cell clusters including ventral forebrain progenitors and early cortical interneurons (*NKX2.1*[+], *DLX2*[+], *GAD1*[+]), cycling progenitors (*TOP2A*[+]), astrocytes (*S100A10*[+]), and a small cluster of radial glia (*SOX9*[+]); as expected due to the previous fusion into hFA, we identified a cluster of cortical glutamatergic neurons (*STMN2*[+], *NEUROD2*[+]; *Figure 3B*, *Figure 3—figure supplement 1A*). For hCOs non-exposed (n=6786 cells) and exposed to hypoxia (n=5781 cells), we also identified five

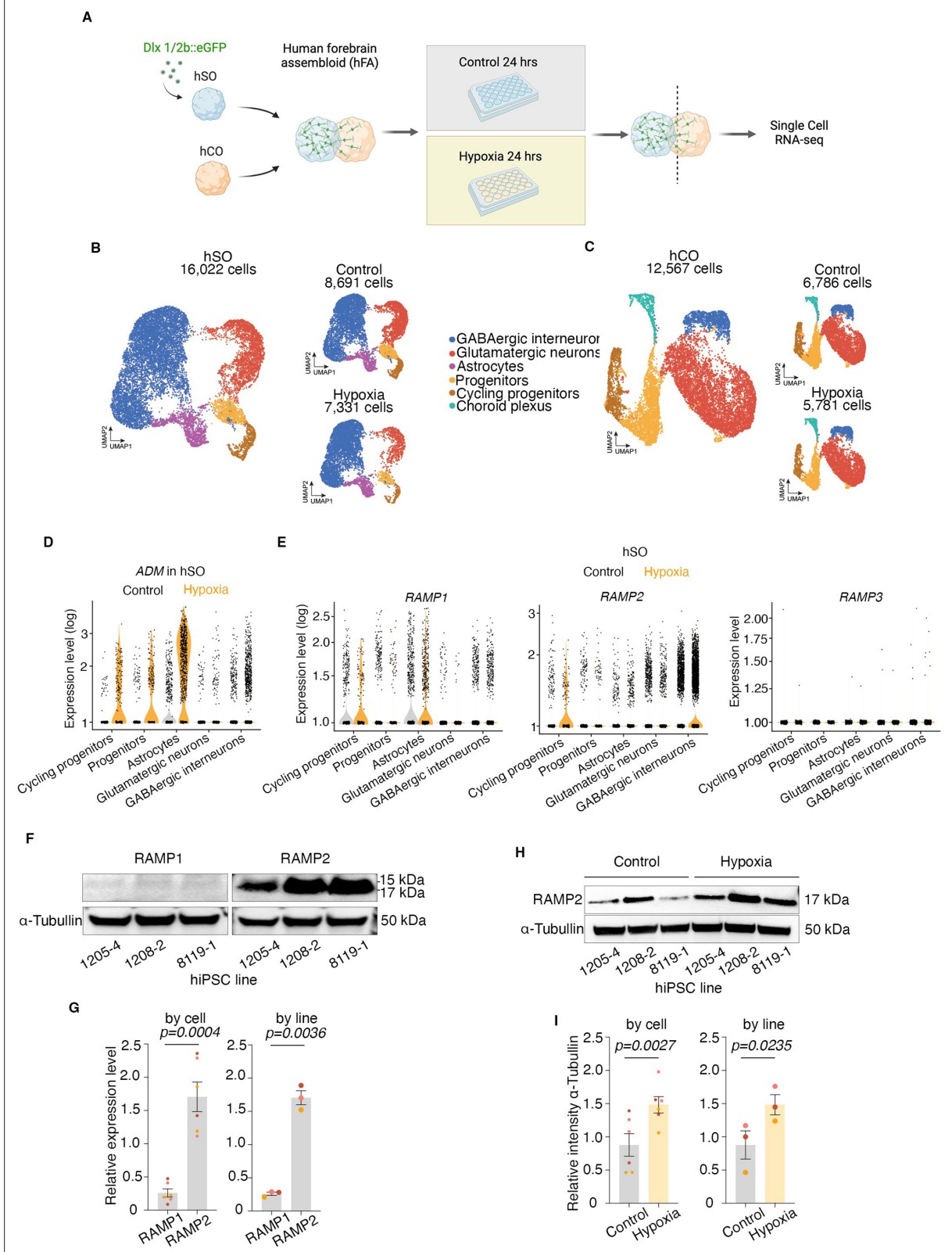

**Figure 3.** Single-cell transcriptional profiling in control and hypoxia-exposed hSO and hCO. Schematic of single-cell RNA-seq of hCO and hSO from hFA; Created with BioRender.com. (**B**) UMAP visualization of the resolved single-cell RNA-seq data of hSO with assignment of main cell clusters, with control and hypoxia exposure shown by condition (*n[total]*=16,022 cells, *n[control]*=8691 cells, *n[hypoxia]*=7331 cells); (**C**) UMAP visualization of the resolved single-cell RNA-seq data of hCO with assignment of main cell clusters, with control and hypoxia exposure shown by condition

*Figure 3 continued on next page*

*Figure 3 continued*

(*n[total]*=12,567 cells, *n[control]*=6786 cells, *n[hypoxia]*=5781 cells); (**D**) Single cell gene expression level (log) of adrenomedullin (*ADM*) in main cell clusters of hSO under control and hypoxia conditions; (**E**) Single-cell gene expression level (log) of *RAMP1* in main cell clusters of hSO under control and hypoxia conditions; Single-cell gene expression level (log) of *RAMP2* in main cell clusters of hSO under control and hypoxia conditions; Single-cell gene expression level (log) of *RAMP3* in main cell clusters of hSO under control and hypoxia conditions; (**F**) (**Left**) Representative blot for RAMP1 protein expression in control conditions, (**Right**) Representative blot for RAMP2 protein expression in control conditions; normalized to α-Tubulin; (**G**) Quantification of RAMP1 and RAMP2 protein expression by (**Left**) individual hSO sample (*two*-tailed paired test, p=0.0004) and (**Right**) by hiPSC line (*two*-tailed paired test, p=0.0036) in control conditions; (**H**) Representative blots for RAMP2 protein expression changes in control and hypoxia conditions; (**I**) Quantification of RAMP2 protein expression (**Left**) by individual hSO sample (*two*-tailed paired test, p=0.0027) and (**Right**) by hiPSC line (*two*-tailed paired test, p=0.0235) in control and hypoxia conditions. Bar charts: mean ± s.e.m.; Different dot colors represent individual hiPSC lines.

The online version of this article includes the following source data and figure supplement(s) for figure 3:

**Source data 1.** Original files for western blot analysis are shown in *Figure 3F and H*.

**Source data 2.** PDF file containing original western blots *Figure 3F and H*, including the relevant bands, iPSC lines, and experimental conditions.

**Figure supplement 1.** Quality control and robustness analyses for scRNA-sequencing data.

cellular clusters consisting of dorsal forebrain progenitors (*SOX2*⁺), cycling progenitors (*TOP2A*⁺), cortical glutamatergic neurons (*STMN2*⁺, *NEUROD2*⁺), and a small cluster of choroid plexus cells (*TTR*⁺); as expected due to the migration of interneurons from the hSO side, we identified a cluster of cortical interneurons (*NKX2.1*⁺, *GAD1*⁺; *Figure 3C*, *Figure 3—figure supplement 1B*). All these cell clusters from hSOs and hCOs demonstrated increased expression of hypoxia-related genes (PDK1, PFKP; *Figure 3—figure supplement 1C, D*). Moreover, we performed subcluster analyses of GABAergic interneurons on hSO and identified that they primarily express markers for MGE and CGE including *NKX2.1*, *LHX6*, somatostatin (*SST*) or calretinin (*CALB2*), less calbindin (*CALB1*), and minimal parvalbumin (*PVALB*), and we validated these findings and performed immunocytochemistry for the following interneuron subtypes: somatostatin (*SST*), calbindin (*CALB*), and calretinin (*CALB2*; *Figure 3—figure supplement 1E, F*).

Next, we assessed the expression of *ADM* in individual cell types. We found that under hypoxic conditions, the expression level of *ADM* increases in all cell types, but more so in ventral neural progenitors and astrocytes within hSOs, and in dorsal neural progenitors and choroid plexus cells in hCOs (*Figure 3D*, *Figure 3—figure supplement 1G*).

Lastly, we evaluated the cell-type-specific expression of *RAMP1, RAMP2,* and *RAMP3,* the known receptors for ADM peptide (*Kuwasako et al., 2011*; *Kuwasako et al., 2004*). Since the previous literature suggests RAMP2 is the main receptor for ADM, we checked the expression of the *RAMP1-3* receptors in control conditions and found that *RAMP2* is indeed the most expressed receptor. Interestingly, we found this receptor is preferentially expressed by interneurons (hSOs) and glutamatergic neurons (hCOs) when compared to other cell types and its transcription is increased in hypoxia (*Figure 3E*, *Figure 3—figure supplement 1H*).

To validate the findings about the higher expression of RAMP2 in control conditions at a protein level, we performed western blot analyses for RAMP1 and RAMP2 in hSOs in control conditions and found that, indeed, RAMP2 was significantly more expressed at baseline than RAMP1 by individual cell (two-tailed paired *t*-test, p=0.0004) and by hiPSC line (two-tailed paired *t*-test, p=0.0036; *Figure 3F and G*). To validate the findings about the increase in RAMP2 expression in hypoxia in hSOs, we checked for changes in the expression of RAMP2 at a protein level and identified a significant increase when analyzed by individual cell (two-tailed paired *t*-test, p=0.0027) and by hiPSC line (two-tailed paired *t*-test, p=0.0235; *Figure 3H and I*). These experiments were performed in three separate biological replicates of hSOs, each derived from four hiPSC lines in one differentiation experiment. Details by hiPSC line are presented in *Supplementary file 1*.

Together, these results provide the first cell type-specific characterization of the expression of ADM and its receptors in the brain in control and hypoxic conditions. Importantly, we identify complementary responses of different cell types, with neurons increasing the expression of the receptors for ADM, while astrocytes, choroid plexus, and progenitors increasing the expression of ADM, suggesting a possible evolutionary adaptation mechanism of cells to protect neurons during hypoxia as an environmental stressor.

## Administration of exogenous ADM during exposure to hypoxia rescues the migration deficits of human cortical interneurons

Based on recent reports about the protective role of exogenous ADM administration in various disease processes (*Zhang et al., 2009*; *Pedreño et al., 2014*; *Gao et al., 2018*; *Ashizuka et al., 2016*; *Clementi et al., 1998*), and in the light of our findings described above, we tested the rescue potential of exogenous ADM administration under hypoxic conditions.

We exposed hFAs to hypoxia in the presence of 0.5 µM human ADM (Anaspec, AS-60447; *Figure 4A*) and imaged individual migrating cortical interneurons (*Figure 4B*). This dose was chosen based on previous reports from in vitro cultures (*Temmesfeld-Wollbrück et al., 2009*). We again observed a significant decrease in the number of saltations under hypoxic conditions when analyzed by individual cells (paired Wilcoxon test, p<0.0001) and by hiPSC line (two-tailed paired *t*-test, p=0.02), and identified rescue of the number of saltations in hypoxia +ADM conditions when analyzed by individual cells (paired Wilcoxon test, p=0.3) and by hiPSC line (two-tailed paired *t*-test, p=0.9; *Figure 4C*).

To check whether exogenous ADM peptide changes the migration of cortical interneurons under control conditions, or whether this is hypoxia-specific, we analyzed the number of saltations of cortical interneurons under control conditions for a total of 48 hr: 24 hr under control conditions (0–24 hr), followed by 24 hr in control +ADM (24–48 hr). We identified no difference in the total number of saltations/24 hr when analyzed by individual cells (paired Wilcoxon test, p=0.23), and hiPSC line (two-tailed paired *t*-test, p=0.77; *Figure 4D*), suggesting ADM is not a main mechanism for migration under baseline conditions. These results align with the data about the low baseline expression of ADM (*Figure 2G*) in control conditions, and its known increase as an acute phase reactant in hypoxia and inflammation (*Ishiyama et al., 2023*; *Valenzuela-Sánchez et al., 2016*; *Solé-Ribalta et al., 2022*).

To verify that the saltation rescue is directly linked to the administration of exogenous ADM, we next performed two types of experiments.

First, we disrupted the disulfide bond between Cys16 and Cys21, which has been shown to be important for ADMs' biological activity (*Fischer et al., 2019*), by denaturing and alkylating it using iodoacetamide (*Figure 4E*, *Figure 4—figure supplement 1A*). Next, we repeated the hypoxia experiments in the presence of 0.5 µM human denatured ADM (dADM) and demonstrated that it does not rescue the saltation deficit under hypoxic conditions, when analyzed by individual cells (paired Wilcoxon test, p<0.0001) and by hiPSC line (paired *t*-test, p=0.04; *Figure 4F*).

Second, we performed pharmacological blocking of RAMP2, the primary recognized receptor for ADM in the literature and in our data. For this, we added 10 µM ADM fragment 22–52 (ADM$_{22-52}$; Cayman Chemical Company 24892) to the hypoxic media already supplemented with exogenous ADM (*Figure 4G*). We validated the saltation deficit in hypoxia and the rescue by exogenous ADM and found that the addition of ADM$_{22-52}$ to the hypoxia +ADM condition was sufficient to prevent the saltation rescue when analyzed in individual cells (paired Wilcoxon test, p<0.0001) and by hiPSC line (paired *t*-test, p=0.003; *Figure 4H*).

## Migration deficits persist in the acute phases of reoxygenation, but administration of exogenous ADM during hypoxia provides partial protection

To investigate whether migration of interneurons is restored upon reoxygenation, we quantified the saltations of individual migrating cortical interneurons for a total of 72 hr: 24 hr in control (0–24 hr), 24 hr in hypoxia (24–48 hr), followed by 24 hr in reoxygenation (48–72 hr). Similar to our experiments from *Figures 1 and 4*, we observed a ~53% decrease in saltations in hypoxia compared to control (Friedman test, p<0.0001), and a 65% decrease in the first 24 hr of reoxygenation compared to control (Friedman test, p<0.0001) when analyzed by individual cells (*Figure 4—figure supplement 1B*). In contrast, the presence of exogenous ADM during hypoxia exposure (not during reoxygenation) fully rescued the migration in hypoxia as previously shown in *Figure 3* and provided partial protection in the first 24 hr after reoxygenation by decreasing the migration of interneurons by 35% compared to control when analyzed by individual cells (Friedman test, p=0.05; *Figure 4—figure supplement 1C*) and by hiPSC line (one-way ANOVA, p=0.84; *Figure 4—figure supplement 1C*); again, this is in contrast to the hypoxia conditions without ADM that see a 65% decrease in saltations during reoxygenation.

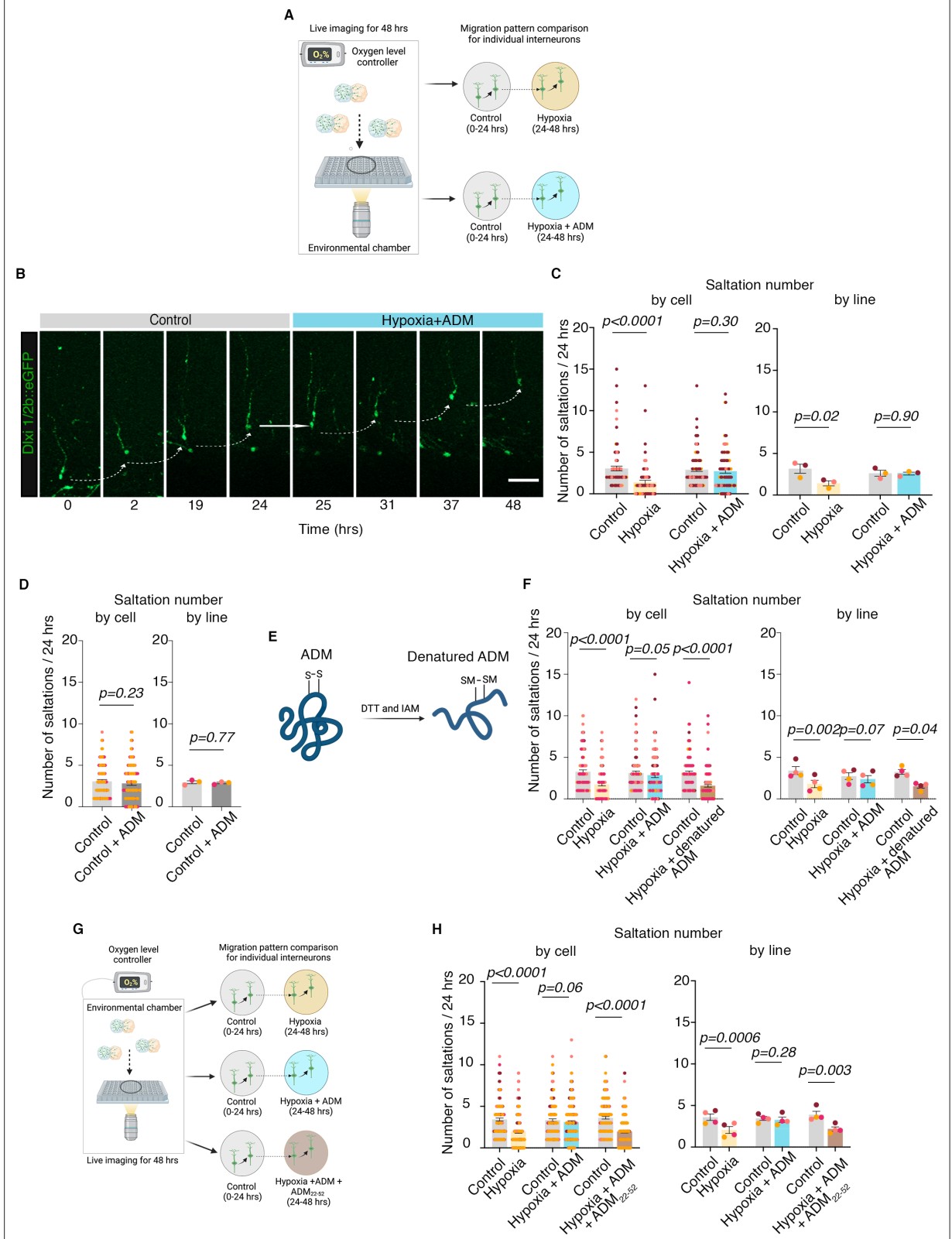

**Figure 4.** Exogenous administration of ADM peptide rescues the migration defects in hypoxia-exposed human cortical interneurons in an ex vivo model using human prenatal cerebral cortex at mid-gestation. (**A**) Schematic of experimental design for pharmacological rescue experiments using ADM; 0.5 μM ADM was added to the media at the beginning of hypoxia exposure; Created with BioRender.com. (**B**) Example of migration pattern for one interneuron in control versus hypoxia +ADM conditions; (**C**) (**Left**) Quantification of saltation number/24 hr in control versus hypoxia conditions

*Figure 4 continued on next page*

*Figure 4 continued*

(paired Wilcoxon test, p<0.0001) and in control versus hypoxia +ADM by individual cells (paired Wilcoxon test, p=0.3); (**Right**) Quantification of saltation number/24 hr in control versus hypoxia conditions (two-tailed paired *t*-test, p=0.02) and in control versus hypoxia +ADM conditions by hiPSC line (two-tailed paired *t*-test, p=0.90); (**D**) (**Left**) Quantification of saltation number/24 hr in control versus control +ADM conditions by individual cells (paired Wilcoxon test, p=0.23); (**Right**) Quantification of saltation number/24 hr in control versus control +ADM conditions by hiPSC line (two-tailed paired *t*-test, p=0.77); (**E**) Schematic of denaturing procedure for ADM, including reduction and alkylation using dithiothreitol (*DTT*) and iodoacetamide (*IAM*), resulting in carbamidomethylated (*CAM*) cysteines at Cys16 and Cys21; Created with BioRender.com. (**F**) (**Left**) Quantification by individual cells of saltation number/24 hr in control versus hypoxia (paired Wilcoxon test, p<0.0001), control versus hypoxia +ADM (paired Wilcoxon test, p=0.05) and control versus hypoxia +denatured ADM (paired Wilcoxon test, p<0.0001); (**Right**) Quantification by hiPSC line of saltation number/24 hr in control versus hypoxia (two-tailed paired *t*-test, p=0.002), control versus hypoxia +ADM (two-tailed paired *t*-test, p=0.07) and control versus hypoxia +denatured ADM (two-tailed paired *t*-test, p=0.04); (**G**) Schematic of experimental design for pharmacological rescue experiments using $ADM_{22-52}$ receptor blocker; Created with BioRender.com. (**H**) (**Left**) Quantification by individual cells of saltation number/24 hr in control versus hypoxia conditions (paired Wilcoxon test, p<0.0001), control versus hypoxia +ADM (paired Wilcoxon test, p=0.06), and control versus hypoxia +ADM + $ADM_{22-52}$ conditions (paired Wilcoxon test, p<0.0001); (**Right**) Quantification of saltation number/24 hr by hiPSC line in control versus hypoxia conditions (paired t-test, p=0.0006), control versus hypoxia +ADM (paired t-test, p=0.28), and control versus hypoxia +ADM + $ADM_{22-52}$ conditions (two-tailed paired t-test, p=0.003). Bar charts: mean ± s.e.m.; scale bar: 50 μm.

The online version of this article includes the following figure supplement(s) for figure 4:

**Figure supplement 1.** Additional data to support the inactivation of ADM and the effects of reoxygenation on migration.

While not the primary focus of the current study because reoxygenation insult has a different biological mechanism of injury (*Datta et al., 2020*; *Kalogeris et al., 2012*) and in-depth analyses would require a different experimental setup, these results reinforce clinical and preclinical data from animal models that reoxygenation injury is important to understand for therapeutic intervention, and that organoid models are able to mimic complex in vivo pathophysiology. Details on the number of cells, experiments, and number of hiPSC lines used for each experiment are presented in *Supplementary file 1*.

## Ex vivo model using developing human brain tissue confirms the migration deficits and rescue by ADM

To verify our findings in an independent human model, we used ex vivo developing human brain tissue at ~20 PCW and followed a similar experimental design used for hFA (*Figure 5A*). Using previously published scRNA-sequencing data from mid-gestation human fetal tissue, we performed subcluster analyses for GABAergic interneurons (*Valenzuela-Sánchez et al., 2016*; *Ashizuka et al., 2016*). We identified three subclusters. One small subcluster expressed doublets and was excluded from subsequent analyses (*Figure 5—figure supplement 1A*; *Clementi et al., 1998*). The remaining two subclusters were found to express MGE- and CGE-derived markers consisting of NKX2.1, LHX6, CALB2, SST, but minimal CALB1 and *PVALB* similar to the ones in hFAs (*Figure 5—figure supplement 1B*).

First, we sectioned human cerebral cortex sections into 400 μm slices (*Figure 5B*) and cultured them on live imaging-compatible cell culture inserts (*Robinson et al., 2006*) (diameter, 23.1 mm; pore size, 0.4 μm; Falcon, 353090). To visualize migrating cortical interneurons, we fluorescently tagged cells using the same forebrain interneuron cell-type specific lentiviral reporter Dlxi1/2b::eGFP (*Figure 5C*; *Potter et al., 2009*).

To assess whether exposure to hypoxia is sufficient to induce a migration deficit in ex vivo human cortical interneurons as observed in hFA, we employed the live-imaging setup established for hFA and monitored the migration patterns of the Dlxi1/2b::eGFP⁺-tagged interneurons for 24 hr under control conditions (37 °C, 5% $CO_2$ 21% $O_2$). We then transitioned the sections into pre-equilibrated hypoxic media (37 °C, 5% $CO_2$, 94.5% $N_2$, <1% $O_2$) and imaged the same cells for another 24 hr. Similar to hypoxic hSOs, we observed a significant transcriptional increase in *ADM* gene by qPCR (two-tailed unpaired *t*-test, p=0.0005; *Figure 5D*). Following exposure to hypoxia for 24 hr, we identified a ~55% decrease in the number of saltations (paired Wilcoxon test, p<0.0001; *Figure 5E*), but no changes in the average saltation length (paired Wilcoxon test, p=0.88; *Figure 5F*) nor in directionality (Wilcoxon test, p=0.3; *Figure 5G*).

Lastly, we found the addition of 0.5 μM ADM peptide during the hypoxia exposure was sufficient to rescue the saltation numbers (paired Wilcoxon test, p=0.09; *Figure 5H*).

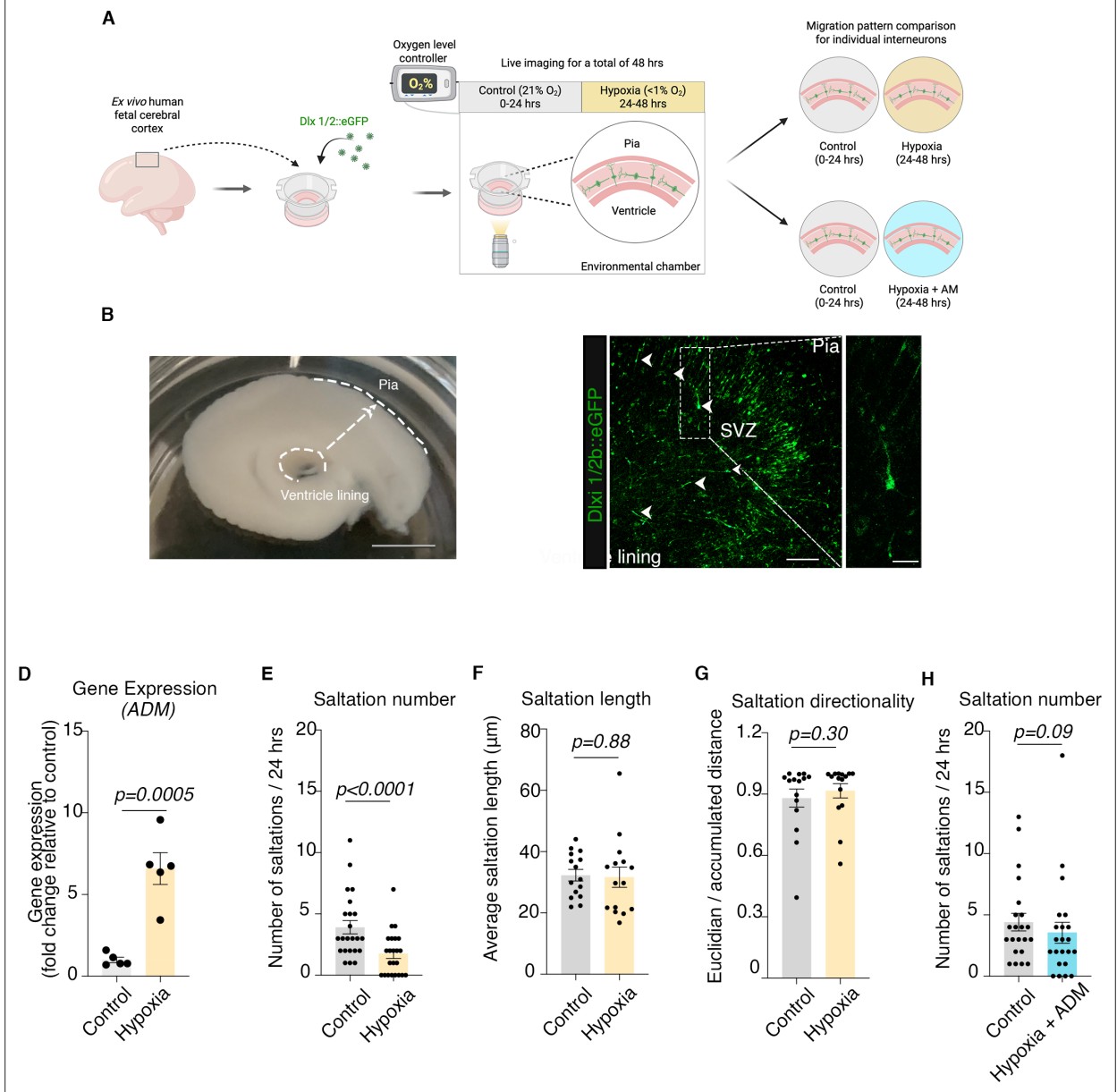

**Figure 5.** Migration defect and rescue by ADM in an ex vivo model using human prenatal cerebral cortex at mid-gestation. (**A**) Schematic illustrating the overall experimental design: sections of ex vivo human cerebral cortex were collected and initially sectioned at ~3 mm and subsequently at 400 μm thickness; sections were transferred onto cell culture membrane inserts suspended in culture media; for visualization, tissue was transfected with Dlxi1/2b::eGFP lentivirus and imaged 7–10 days post infection directly on inserts; GFP-tagged ex vivo human prenatal cortical interneurons were monitored for 24 hr in control conditions and 24 hr in hypoxic conditions in the presence or absence of 0.5 μM ADM; Created with BioRender. com. (**B**) Example of macroscopic view of a 3 mm section of fresh ex vivo human prenatal cerebral cortex; scale bar: 1 cm; (**C**) Representative image of fluorescently tagged cortical interneurons in a section of ex vivo human prenatal cerebral cortex; (**D**) Transcriptional increase of *ADM* gene following 24 hr of exposure to hypoxia of ex vivo human prenatal cerebral cortex samples (two-tailed unpaired *t*-test, p=0.0005); (**E**) Quantification of saltation numbers/24 hr in control versus hypoxia conditions (paired Wilcoxon test, p<0.0001); (**F**) Quantification of average saltation length for control versus hypoxia conditions (paired Wilcoxon test, p=0.88); (**G**) Quantification of directionality of migration for control versus hypoxia conditions (paired Wilcoxon test, p=0.3); (**H**) Quantification of saltation numbers/24 hr in control versus hypoxia +ADM conditions by individual cells (paired Wilcoxon test, p=0.09). Bar charts: mean ± s.e.m.; scale bars: 1 cm, 200 μm and 50 μm.

The online version of this article includes the following figure supplement(s) for figure 5:

**Figure supplement 1.** Additional data to support molecular mechanisms of rescue by ADM.

## Proposed molecular mechanism that contributes to the rescue by exogenous ADM administration

First, we performed EIA analyses focused on cAMP/PKA, AKT/pAKT, and ERK/pERK pathways, which have been previously shown to be modulated by ADM (*Ozcelik et al., 2019*; *Figure 6A*). We found that the concentration of cAMP was not affected by hypoxia but was significantly increased in the hypoxia +ADM conditions, both when analyzed by individual samples (one-way ANOVA test, p=0.0007), and by hiPSC line (one-way ANOVA test, p=0.002; *Figure 6B*). These results were complemented by increased PKA activity in hypoxia +ADM conditions when analyzed by individual samples (one-way ANOVA test, p<0.0001), and by hiPSC line (one-way ANOVA, p<0.0001; *Figure 6C*). In contrast, the ratio of pAKT/AKT was significantly decreased in hypoxia samples when analyzed by individual samples (one-way ANOVA, p<0.0001) and by hiPSC line (Kruskal-Wallis test, p=0.01), but was not rescued in hypoxia +ADM (*Figure 6D*). These findings were similar for the ratio of pERK/ERK in individual samples (one-way ANOVA test, p=0.001) and by hiPSC line (one-way ANOVA, p=0.02; *Figure 6E*). Lastly, we quantified the ratio of pCREB/CREB, which is downstream of all these pathways. We found that hypoxia induced a significant decrease in the ratio of pCREB/CREB when compared to control when analyzed by individual samples (one-way ANOVA test, p=0.003) and by hiPSC line (p=0.048). However, this ratio was restored in hypoxia +ADM conditions when analyzed by individual samples (one-way ANOVA test, p=0.27) and by hiPSC lines (one-way ANOVA, p=0.6; *Figure 6F*). These results suggest the cAMP/PKA/pCREB pathway is modulated by exogenous ADM administration in hypoxia in hSOs.

Previous reports demonstrated that increased ratio of pCREB/CREB is directly involved in increasing the expression of GABA$_A$ receptors (*Figure 6G*; *Fukuchi et al., 2014*; *Birey et al., 2022*), which in turn, are well-documented to facilitate interneuron migration (*Luhmann et al., 2015*). To check whether the expression of GABA$_A$ receptors is affected under hypoxic conditions and rescued by ADM, we quantified the expression of several GABA$_A$ receptor subunits in the presence or absence of ADM during hypoxia exposure. We specifically focused on the most common pentameric combination, α1β2γ2, encoded by *GABRA1*, *GABRB2*, and *GABRG2* (*Connolly et al., 1996*; *Figure 6G*). We found a significant decrease in expression in hypoxia (one-way ANOVA test; control vs hypoxia: p=0.002) and rescue by ADM for *GABRA1* (one-way ANOVA test; p=0.17); no significant change in expression in *GABRB2* (Kruskal-Wallis test; control versus hypoxia: p=0.08; control versus hypoxia +ADM: p=0.25); a significant decrease in expression in hypoxia (one-way ANOVA test; control versus hypoxia: p=0.007) and rescue by ADM for *GABRG2* (one-way ANOVA test; p=0.23; *Figure 6H*). Moreover, we analyzed other receptor subunits and found mild but still significant decrease in hypoxia and rescue by ADM for receptors *GABRB3* (Kruskal-Wallis test; control versus hypoxia: p=0.02; control versus hypoxia +ADM: p=0.05) and *GABRG3* (one-way ANOVA test; control versus hypoxia: p=0.01; control versus hypoxia +ADM: p=0.1); no change in expression under hypoxic conditions for receptors *GABRA2* (one-way ANOVA; control versus hypoxia: p=0.8; control versus hypoxia +ADM: p=0.75; *Figure 6—figure supplement 1A*).

Separately, we investigated the expression of *CXCR4*, *CXCR7,* and *CXCL12,* as some of the best-known chemokines/chemokine receptors involved in interneuron migration (*Wang et al., 2011*; *Toudji et al., 2023*). Interestingly, we found a significant increase of *CXCR4* in hypoxia but no rescue by ADM (one-way ANOVA test; control versus hypoxia: p=0.0005; control versus hypoxia +ADM: p=0.0031), a significant decrease of *CXCR7* in hypoxia but no rescue by ADM (one-way ANOVA test; control versus hypoxia: p=0.026; control versus hypoxia +ADM: p=0.043), and a significant decrease of *CXCL12* in hypoxia but no rescue by ADM (one-way ANOVA test; control versus hypoxia: p=0.024; control versus hypoxia +ADM: p=0.0039; *Figure 6I*).

Overall, these results suggest that rescue of GABA$_A$ receptor expression is sufficient to restore migration patterns in hypoxic cortical interneurons, even in the absence of rescue of cytokine receptors.

A graphical summary of the proposed mechanims of protection by ADM in presented in *Figure 7A*.

## Discussion

Delayed or reduced migration of inhibitory interneurons into the developing cortex can have profound consequences for neural circuit formation and function. Proper timing and integration of GABAergic interneurons are essential for establishing the excitatory–inhibitory (E/I) balance that regulates cortical

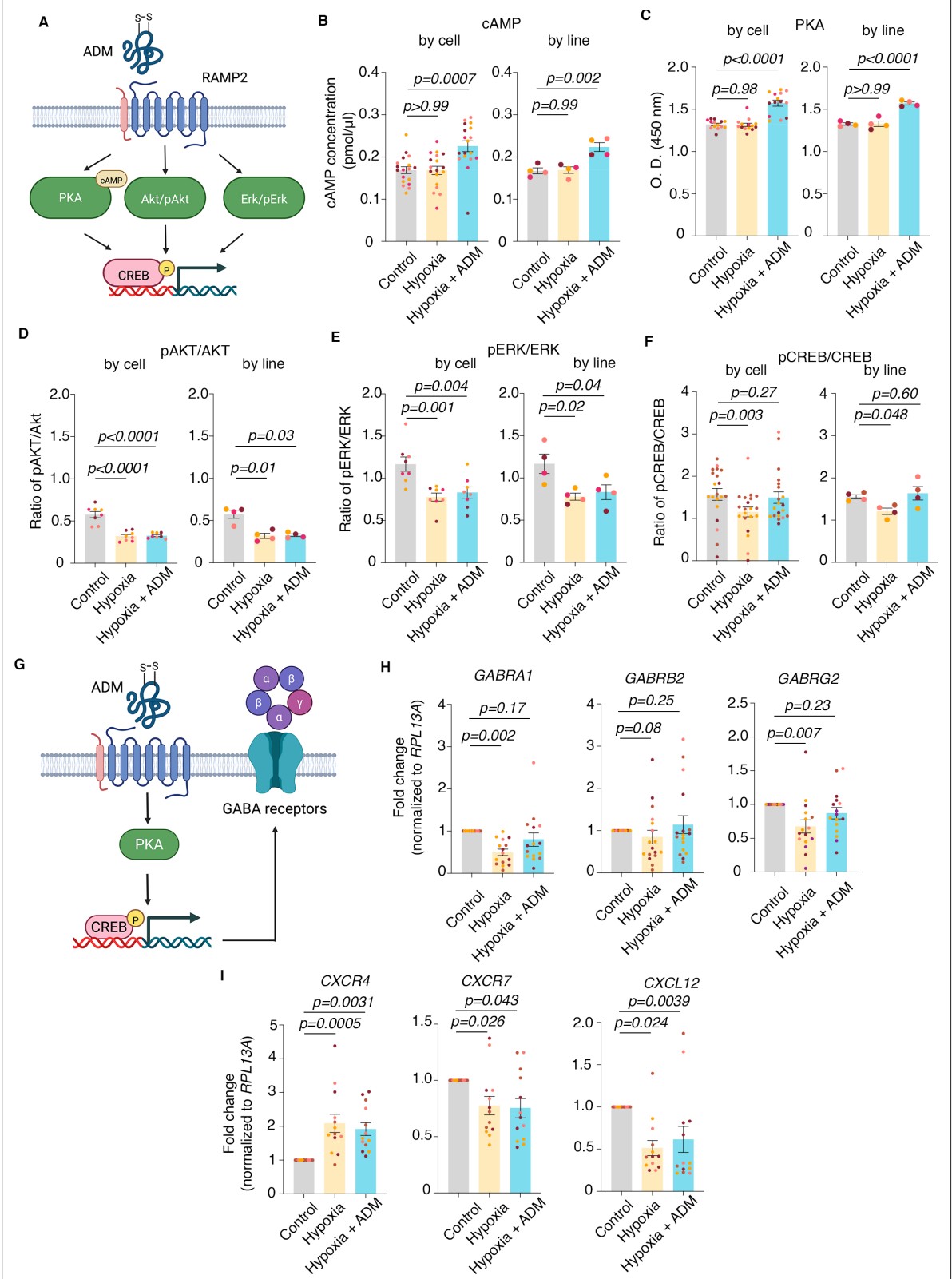

**Figure 6.** Molecular mechanism of rescue by ADM. (**A**) Schematic of the previously reported main molecular pathways modulated by ADM; Created with BioRender.com. (**B**) (**Left**) Quantification of cAMP concentration (pmol/μL) by individual hSOs in control versus hypoxia (one-way ANOVA, p>0.99) and, control versus hypoxia +ADM (one-way ANOVA, p=0.0007), (**Right**) Quantification of cAMP concentration (pmol/μL) by hiPSC line in control versus hypoxia (one-way ANOVA, p=0.99) and, control versus hypoxia +ADM (one-way ANOVA, p=0.002); (**C**) (**Left**) Quantification of PKA activity (O.D.

*Figure 6 continued on next page*

*Figure 6 continued*

450 nm) by individual hSOs in control versus hypoxia (one-way ANOVA test, p=0.98) and control versus hypoxia +ADM (one-way ANOVA, p<0.0001); (**Right**) Quantification of PKA activity (O.D. 450 nm) by hiPSC line in control versus hypoxia (one-way ANOVA, p>0.99) and control versus hypoxia +ADM (one-way ANOVA, p<0.0001); (**D**) (**Left**) Quantification of pAKT/AKT by individual hSOs in control versus hypoxia (one-way ANOVA, p<0.0001) and control versus hypoxia +ADM (one-way ANOVA, p<0.0001); (**Right**) Quantification of pAKT/AKT by hiPSC line in control versus hypoxia (Kruskal-Wallis test, p=0.01) and control versus hypoxia +ADM (Kruskal-Wallis test, p=0.03); (**E**) (**Left**) Quantification of pERK/ERK by individual hSOs in control versus hypoxia (one-way ANOVA test, p=0.001) and control versus hypoxia +ADM (one-way ANOVA test, p=0.004); (**Right**) Quantification of pERK/ERK by hiPSC line in control versus hypoxia (one-way ANOVA test, p=0.02) and control versus hypoxia +ADM (one-way ANOVA test, p=0.04); (**F**) (**Left**) Quantification of pCREB/CREB by individual hSOs in control versus hypoxia (one-way ANOVA test, p=0.003) and control versus hypoxia +ADM (one-way ANOVA test, p=0.27); (**Right**) Quantification of pCREB/CREB by hiPSC line in control versus hypoxia (one-way ANOVA test, p=0.048) and control versus hypoxia +ADM (one-way ANOVA test, p=0.6); (**G**) Schematic of the proposed molecular pathway activation by exogenous ADM in hSOs, including the most common pentameric structure of the GABA$_A$ receptor; Created with BioRender.com. (**H**) (**Left**) Quantification (by q-PCR) of GABRA1 in hSOs samples in control versus hypoxia conditions (one-way ANOVA test, p=0.002) and control versus hypoxia +ADM (one-way ANOVA test, p=0.17); (**Center**) Quantification (by q-PCR) of GABRB2 in hSOs samples in control versus hypoxia (one-way ANOVA test, p=0.08) and control versus hypoxia +ADM (one-way ANOVA test, p=0.25); (**Right**) Quantification (by q-PCR) of GABRG2 in hSO samples in control versus hypoxia conditions (one-way ANOVA test, p=0.007) and control versus hypoxia +ADM (one-way ANOVA test, p=0.23); (**I**) (**Left**) Quantification (by q-PCR) of CXCR4 in hSOs samples in control versus hypoxia conditions (one-way ANOVA test, p=0.0005) and control versus hypoxia +ADM (one-way ANOVA test, p=0.0031); (**Center**) Quantification (by q-PCR) of CXCR7 in hSOs samples in control versus hypoxia conditions (one-way ANOVA test, p=0.026) and control versus hypoxia +ADM (one-way ANOVA test, p=0.043); (**Right**) Quantification (by q-PCR) of CXCL12 in hSOs samples in control versus hypoxia conditions (one-way ANOVA test, p=0.024) and control versus hypoxia +ADM (one-way ANOVA test, p=0.0039). Bar charts: mean ± s.e.m.; Different dot colors represent individual hiPSC lines.

The online version of this article includes the following figure supplement(s) for figure 6:

**Figure supplement 1.** Additional data to support molecular mechanisms of rescue by ADM.

oscillations, synaptic refinement, and neuronal synchrony (*Kepecs and Fishell, 2014*; *Marín, 2012*). Disruption of interneuron migration can lead to abnormal circuit assembly and hyperexcitability, ultimately impairing synaptic plasticity and cortical information processing (*Anderson et al., 1997*; *Wonders and Anderson, 2006*). Functionally, such alterations have been associated with several neurodevelopmental and neuropsychiatric conditions, including epilepsy, autism spectrum disorder (ASD), and schizophrenia (*Marín, 2016*; *Lewis et al., 2005*).

Disruptions in cortical interneurons' migration from the ganglionic eminences toward the cortex have been long suggested as a contributing factor to the increased risk for neuropsychiatric diseases in hypoxic brain injury of prematurity, because this critical developmental process occurs predominantly during the latter half of pregnancy and is likely to be affected by preterm birth. However, the migration patterns under hypoxic conditions have proved difficult to investigate in animal models, and no data exist for human cortical interneurons.

Given the accumulating evidence demonstrates the power of human cellular models for the study of neurodevelopmental diseases of genetic or environmental etiology, including tuberous sclerosis, Timothy syndrome, 22q11.2 deletion syndrome, ZikV exposure, preterm birth, etc (*Birey et al., 2022*; *Kelley and Pașca, 2022*; *Blair and Bateup, 2020*; *Li et al., 2025*; *Qian et al., 2016*), in this study, we established the first live imaging microscopy platform for the direct investigation of the migration patterns on human cortical interneurons in control and hypoxic conditions, thus overcoming a long-term challenge in the field. The high technical and scientific relevance of this novel platform for disease investigation is further amplified by the concomitant use of two human models consisting of hFAs derived from hiPSCs and ex vivo developing human cortical tissue, an approach that helps mitigate potential confounding factors associated with possible interspecies differences in how brain cells respond to hypoxia and other environmental stressors.

Using these human cellular platforms, we directly demonstrate a substantial saltation deficit of human cortical interneurons exposed to mild hypoxia. Importantly, we provide the first characterization of molecular changes in interneurons in hypoxia, identify exogenous ADM peptide supplementation as an effective pharmacological rescue for this phenotype, and bring evidence that this is, at least partly, mediated by the cAMP/PKA/pCREB pathway and rescue of the expression of specific subunits that form the α1β2γ2 pentameric GABA$_A$ receptor. Moreover, in this study, we provide preliminary evidence for the ongoing migration deficits in the first 24 hr after reoxygenation, suggesting a longer-term effect of a hypoxic event and the possible protective effect of exogenous ADM administration. Lastly, we provide evidence that the endogenous production of ADM is not sufficient for a phenotypic

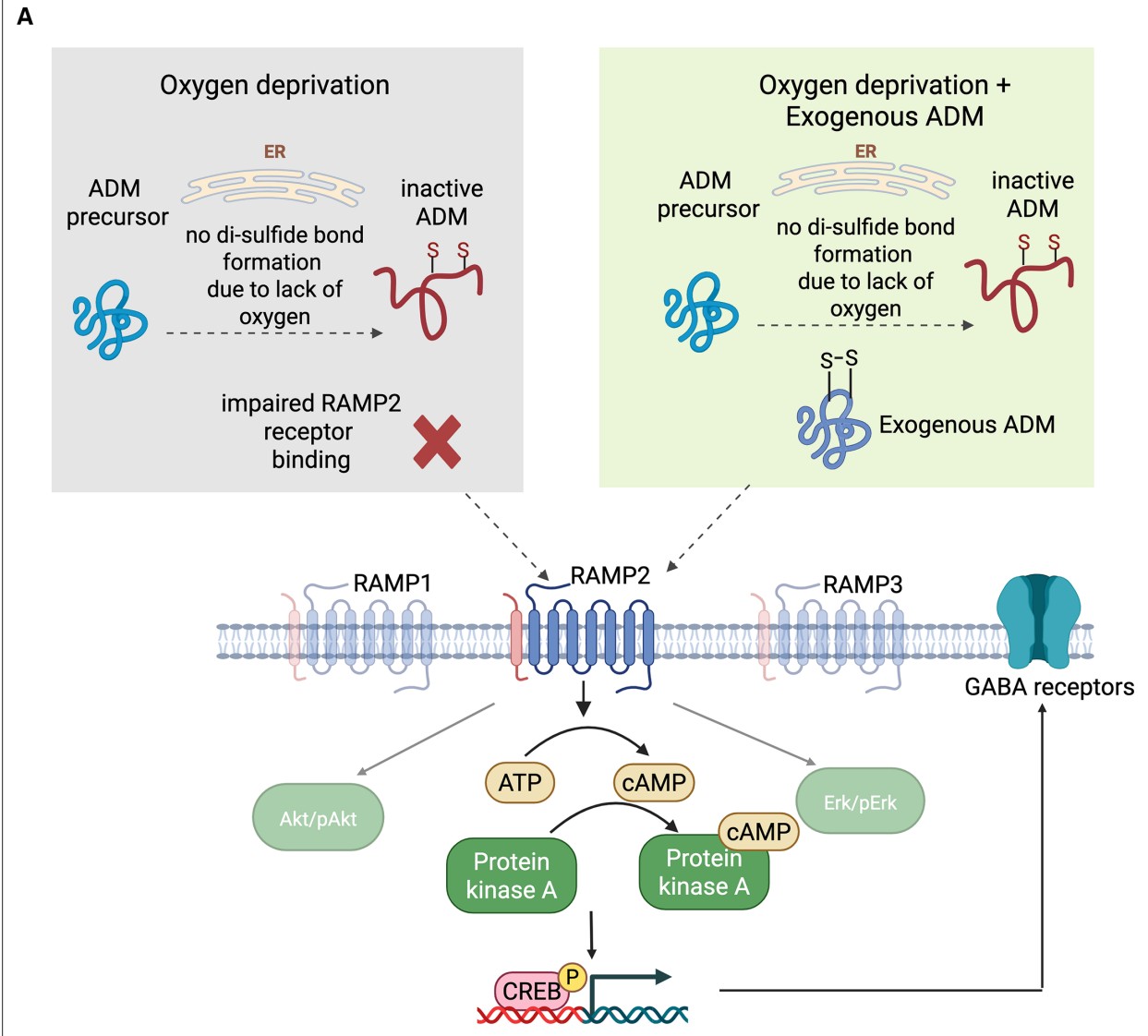

**Figure 7.** Schematic of the overall proposed mechanism of interneuron migration defect rescue by ADM upon hypoxia exposure. (**A**) Based on our findings and existing data from literature, we propose endogenously produced ADM has decreased biological activity by impaired ability to form the necessary disulfide bond in the absence of oxygen in hypoxia. However, exogenous ADM does have biological activity as the disulfide bond is present, and thus it binds efficiently to its receptors, especially RAMP2. This binding initiates an activation of the cAMP/PKA/pCREB pathway, which in turn restores the expression of GABA_A receptors and rescues the migration. Created with BioRender.com.

rescue, and we suggest this is due to the decreased biological activity of endogenously produced ADM, due to inability to efficiently form the disulfide bond in a hypoxic environment. Since exogenous ADM administration is already showing promise in clinical trials for inflammatory bowel disease (***Kita et al., 2022***), this new knowledge could serve as a starting point for the investigation of ADM as a possible future target for therapeutic interventions for hypoxic brain injuries. Lastly, the novel and interesting findings about the cell-type-specific expression of ADM and its receptors on brain cells add important biologic knowledge and will inform future studies focused on the role of ADM in the brain.

Our manuscript has several limitations. First, while hFAs and ex vivo developing human cortical tissue are the 'best next thing' after in vivo human brain, they do remain a model, like all other preclinical models. The migration of interneurons observed in hSO–hCO assembloids recapitulates several key aspects of in vivo cortical development, including directional migration from subpallial to pallial regions, responsiveness to chemotactic gradients, and integration into cortical-like networks. However,

this model remains a simplified representation of the developing human brain. Several environmental components are absent, including vascularization, immune and glial interactions, long-range axonal inputs, and extracellular matrix complexity, all of which shape interneuron migration and maturation in vivo. Future studies should focus on the translational potential of ADM for brain injury of prematurity using large in vivo animal models (e.g. piglet, lambs, non-human primates) which have more similar brain developmental trajectories to humans, allow reliable modeling of preterm birth and multiple complex analyses (e.g. blood draws, neurological monitoring etc), which are not feasible in organoids. Second, while we show ongoing injury after reoxygenation, we do not provide in-depth phenotypic cellular and molecular characterization of this injury. Since reoxygenation injury is an important, yet separate entity in the complex pathophysiology of hypoxic brain injury, future studies should adapt our current experimental design to focus on this question, which will be highly relevant for the future development of ADM and other compounds as therapeutics. Third, this study was specifically focused on studying the migration of cortical interneurons; however, other developmental processes are likely affected, including the proliferation of ventral forebrain progenitors and the functional integration of mature neurons within the neuronal cortical networks. These should be addressed in future studies. Lastly, here we focused on ADM, while excluding the detailed investigation of other molecular pathways and genes that were differentially expressed, including the inflammatory pathway, which is highly relevant for the pathophysiology of hypoxic brain injury of prematurity.

## Methods

### Culture of hiPSCs

The hiPSC lines used in this study were validated using standardized methods we previously described (*Robinson et al., 2006*; *Takano, 2015*; *Pedreño et al., 2014*; *Connolly et al., 1996*). Cultures were regularly tested for, and maintained Mycoplasma free. SNP-array experiments were performed for genomic integrity. The differentiation experiments were performed using four control hiPSC lines derived from fibroblasts harvested from four healthy individuals (two XX and two XY). Informed consent for fibroblast collection was obtained from all individuals, and this process was approved by the Stanford IRB Panel.

### Generation of region-specific organoids

Brain region-specific human cortical organoids (hCO) and human subpallium organoids (hSO) were differentiated using feeder-free grown hiPSCs using validated protocols we previously reported and contributed to developing (*Robinson et al., 2006*; *Takano, 2015*; *Zhang et al., 2009*; *Pedreño et al., 2014*; *Connolly et al., 1996*; *Wang et al., 2011*; *Toudji et al., 2023*; *Kepecs and Fishell, 2014*; *Marín, 2012*). Briefly, hiPSCs were maintained in six-well plates coated with recombinant human vitronectin (VTN-N, Thermo Fisher Scientific, A14700) in Essential 8 medium (Thermo Fisher Scientific, A1517001). For differentiation, hiPSC colonies were lifted from the plates using Accutase (Innovate Cell Technologies, AT-104). Approximately 3 million cells were transferred to each well in AggreWell 800 (STEMCELL Technologies, 34815) and centrifuged at $100 \times g$ for 3 min in hiPSC medium supplemented with ROCK inhibitor Y-27632 (Selleck Chemicals, S1049). After 24 hr, the newly formed organoids were transferred into ultra-low-attachment plastic dishes (Corning, 430293) in Essential 6 medium (Thermo Fisher Scientific, A1516401) supplemented with dorsomorphin (2.5 µM, SigmaAldrich, P5499) and SB-431542 (10 µM, Tocris, 1614) and additional XAV-939 (hCO 0.5 µM, hSO: 2.5 µM Tocris, 3748). This medium was replaced daily for the first five days. On the sixth day in suspension, organoids were transferred to neural medium containing Neurobasal A (Thermo Fisher Scientific, 10888022), B-27 supplement without vitamin A (Thermo Fisher Scientific, 12587010), GlutaMax (2 mM, Thermo Fisher Scientific, 35050061) and penicillin and streptomycin (100 U mL$^{-1}$, Thermo Fisher Scientific, 15140122). For hCO, the neural medium was supplemented with the growth factors EGF (20 ng mL$^{-1}$, R&D Systems, 236-EG) and FGF2 (20 ng mL$^{-1}$, R&D Systems, 233-FB) every day until day 15, then every-other day until day 23. For hSO, we included continued supplementation with XAV-939 (2.5 µM) on days 6–23, and SHH pathway agonist SAG (smoothened agonist; 100 nM; Thermo Fisher Scientific, 566660) on days 12–23. Between days 25 and 43, the neural medium was supplemented with neurotrophic factors BDNF (20 ng mL$^{-1}$, Peprotech, 450–02) and NT3 (20 ng mL$^{-1}$, Peprotech, 450–03), to

promote differentiation of the neural progenitors into neurons in both hCO and hSO; media change was performed every other day. After day 43, hCO and hSO were maintained in neural media.

## Viral labeling of hSO and generation of human forebrain assembloids (hFA)

The viral infection of the hSO for visualization of migrating cortical interneurons was performed as previously described (*Robinson et al., 2006*; *Ozcelik et al., 2019*). Briefly, at approximately 45–55 days in culture, hSO were transferred to a 24-well plate (Corning, 3474) containing 500 µL of neural medium with 10 µL of virus (Lenti-Dlxi1/2b::eGFP; construct reported and applied in reference; *Gao et al., 2018*). After 24 hr, 500 µL of neural medium was added. On day 3, all media was removed and replaced with 1 mL of neurobasal medium. The next day, hSO were transferred into fresh neural media in ultra-low attachment plates. After 5–7 days, hCO and virally infected hSO were used to generate hFA. To do this, one hCO and one hSO were transferred together into a 1.5 mL microcentrifuge Eppendorf tube and maintained in direct contact for 3 days. More than 95% of hCO and hSO fused. Subsequently, hFA were carefully transferred into 24-well ultra-low attachment plates (Corning, 3474), and media changes were performed very gently every two to three days. Lentivirus construct (lenti-Dlxi1/2b::eGFP) was received as a gift from J. L. Rubenstein, and virus was generated by transfecting HEK293T cells with PEI Max (Polysciences, 24765); after 48 hr the media was collected and ultracentrifuged at 17,000 rpm at 6 °C for 1 hr.

## Primary human prenatal cortex processing and viral labeling

Ex vivo developing human cortical tissue at 20 PCW was obtained under a protocol approved by the Research Compliance Office at Stanford University. The tissue was processed within 4 hr after collection using a previously described protocol (*Robinson et al., 2006*; *Ozcelik et al., 2019*; *Connolly et al., 1996*). In brief, cortical tissue was embedded in 4% low melting-point agarose in PBS and cut using a Leica VT1200 Vibratome at 400 µm. The sections were transferred onto cell culture membrane inserts (diameter, 23.1 mm; pore size, 0.4 µm; Falcon, 353090) and incubated in culture media (66% BME, 25% Hanks, 5% FBS, 1% N-2, 1% penicillin, streptomycin, and glutamine; all from Invitrogen) and 0.66% D-(+)-Glucose (Sigma) at 5% $CO_2$, 37 °C with the Dlxi1/2b::eGFP lentivirus for 24 hr. Sections were then transferred to fresh cell culture media, and half media changes were performed every other day. After ~5 days in culture, Dlxi1/2b::eGFP + cells could be detected and imaging was performed within 7–10 days in culture.

## Live imaging and analyses of migration patterns of Dlxi1/2b::eGFP-tagged cortical interneurons

For imaging, hFA were transferred intact into a 96–well plate (glass-bottom, Cellvis, Catalog #: P96-0N) in 300 µL of neural media. Imaging was performed under environmentally controlled conditions (37 °C, 5% $CO_2$, 21% $O_2$) using a confocal microscope with a motorized stage (Zeiss LSM 980) and equipped with a cage incubator for temperature, gas (including $O_2$ levels), and humidity control (Okolab Bold Line). During each recording session, we imaged up to 12 hFA at ×10 magnification, at a depth of 350–400 µm, and at a rate of 20 min/frame. The imaging field was focused on the hCO side of the fusion.

Samples were imaged for 24 hr under control conditions (37 °C, 5% $CO_2$, 21% $O_2$). To induce acute hypoxia, media was carefully removed and replaced with pre-equilibrated hypoxic media. All procedures were performed in the controlled hypoxic environment of the HypoxyLab (Oxford Optronix). The field of view was readjusted to capture the previous region of interest, and the same cells were imaged for an additional 24 hr while maintaining a hypoxic environment (37 °C, 5% $CO_2$, <1% $O_2$). For pharmacological rescue experiments, the pre-equilibrated hypoxic media containing the peptide ADM (0.5 µM, Anaspec, AS-60447)+/-10 uM $ADM_{22-52}$ (the RAMP2 receptor blocker [Cayman Chemical Company, cat. no. 24892]), were added to the hFA cell culture media during hypoxia exposure.

The imaging of migration of ex vivo human developing cortical interneurons was done with the same settings as described above, with the exception of image depth being 150–200 µm. Slices were kept on the cell culture inserts during imaging.

Post-acquisition analyses of cell mobility were performed using ImageJ (v:2.00-rc-69/1.52 n). Individual cells were identified and followed both in control and hypoxic conditions. Analyses were

performed on all cells that had at least one saltation within the first 24 hrs of imaging (control conditions) and remained visible within the field of view until the end of the imaging period (up to 72 hrs). Paired analyses of number of saltations and saltation length were performed as described in the main text. To estimate the length of individual saltations, we manually tracked and identified the swelling of the soma of Dlxi1/2b::eGFP tagged cortical interneurons before and after saltation and measured the distance (in μm) from the initial position to the new position. The Z-stack alignment present in Zen 3.1 Blue Edition (Zeiss) was used to correct for minor drifts during imaging. To assess changes in directionality of the movement, we extracted the x and y coordinates of each cell per frame and time using the Manual Tracking plugin (ImageJ). The Chemotaxis & Migration Tool (Ibidi) was then used to calculate the Accumulated (A) and Euclidean (E) distances traveled per cell over time. Data is presented by calculating the E/A ratio. Only cells which had at least one saltation within each of the conditions were included in this analysis.

## Hypoxia treatment of organoids and primary tissues for in vitro assays

Organoids or developing human cortical tissue used for in vitro hypoxia experiments were transferred to culture medium pre-equilibrated in a hypoxic C-chamber (Biospherix) or HypoxyLab (Oxford). Hypoxic C-chamber was connected to a premixed gas tank containing 5% $CO_2$ and balancing $N_2$. Oxygen level in the C-chamber was controlled and monitored using a Proox 110 Compact Oxygen Controller to reach <1% $O_2$. HypoxyLab was connected to $O_2$, $N_2$, and $CO_2$ gas tanks and built-in gas sensors control each gas level in real-time to achieve 30 mmHg with 5% $CO_2$. After 24 hrs treatment by hypoxia, organoids or primary tissues were collected immediately for assays.

## Oxygen measurements in control and hypoxic cell culture media

The $O_2$ tension of the media was measured using the oxygen optical microsensor OXB50 (50 μm, PyroScience) attached to a fiber-optic multi-analyte meter (FireStingO$_2$, PyroScience), as previously described (*Fukuchi et al., 2014*; *Marín, 2012*).

## RNA sequencing of hSO

At 46 days in culture, three hSO from four individual hiPSC lines were exposed to hypoxic conditions for 12 and 24 hr using a hypoxia C-chamber as described above. For acute induction of hypoxia, the media was previously equilibrated overnight at <1% $O_2$, 5% $CO_2$, 37 °C with a resulting $PO_2$ of 25–30 mmHg. After 12 hr and 24 hr of hypoxic exposure, hSO were immediately collected and snap-frozen in dry ice for analyses. Control hSO maintained in regular incubator conditions was collected from each hiPSC line at each experimental timepoint. Additionally, extra hSO exposed to 24 hr of hypoxia was transferred back to baseline conditions and reoxygenated for 72 hr. At this time point, samples were collected, matched with control hSO, and snap-frozen on dry ice. All samples were kept at –80 °C until they were processed for RNA-sequencing.

## RNA sequencing analyses

mRNA was isolated using the RNeasy Mini kit (Qiagen) and sequence libraries were prepared by Admera Health (https://www.admerahealth.com) using the KAPA Hyper Prep Stranded RNA-Seq with RiboErase. Sequencing was performed on the Illumina HiSeq 400 system with paired-end 150 base pair reads. Samples averaged 40 million reads, each representing 20 x genome coverage. Trimmed RNA-Seq reads were aligned to the Ensembl GRCh38 human genome reference using STAR v2.7.3 in gene annotation mode (*Lim et al., 2018*). Alignment and RNA-Seq quality control metrics were generated with Picard (v 2.21.1). Principal component analysis (PCA) of the quality control matrices did not indicate any sample outliers.

Gene-level counts were filtered to remove lowly expressed genes which did not have at least 1 count per million in at least 4 out of the total 24 samples; this resulted in 17,777 remaining genes. We noticed a slight GC content bias across samples, so gene counts were further normalized by GC content, gene length, and sequencing depth using the CQN R package v1.36.0 (*Luhmann et al., 2015*). The resulting estimation parameters were provided to the negative binomial generalized linear model while performing the differential expression (DEG) analysis. This was carried out using DESeq2 v1.30.1 to identify significant genes between hypoxic conditions at each time point while also accounting for unwanted variation due to cell line (*Wang et al., 2011*). The resulting P values were

corrected for multiple comparisons using the Benjamini-Hochberg method. Genes were considered significant at FDR ≤0.05 and absolute fold change >1.5 (log2FC of 0.6). In total, we identified 734 DEX genes as a consequence of the hypoxic conditions at 12 hr, 985 genes at 24 hr, and only 21 genes at 72 hr after reoxygenation (*Figure 2*). We generated the dendrogram in *Figure 2B* using expression data from all 24 samples, narrowed to the union of the genes differentially expressed between hypoxic and control conditions at 12 and 24 hr or 72 hr after reoxygenation (n=1473). Hierarchical clustering, using complete linkage of Euclidean distance, determined the order of the dendrogram. Z-score normalized expression values are depicted on a continuous scale from lower values (purple) to higher values (orange). The notable reduction of DEG at 72 hr after reoxygenation reflects what we visually see in the PCA of the normalized gene expression, in that the control and hypoxia conditions cluster together at 72 hr post reoxygenation (not shown). Indeed, hierarchical clustering using the Ward's criterion on the first three principal components separates hypoxic samples at 12 and 24 hr from the control 12 and 24 hr samples as well as the reoxygenated samples (*Figure 2—figure supplement 1B*).

To gain insight about putatively dysregulated biological pathways, we conducted gene set enrichment analysis that was computed using the fgsea R package v1.18.0 where all genes were ranked by their absolute $\log_2$ fold change (*Toudji et al., 2023*). We evaluated the Molecular Signatures Database "'allmark' gene sets and supplemented the WU_cell migration set based on its presence in a preliminary gene ontology enrichment. The top 10 significantly enriched gene sets (padj <0.05) that occur across all timepoints are shown in *Figure 2D*. Interested in discovering which genes had the largest change in response between 12 hr and 24 hr, we identified genes significant at both timepoints and ranked them by absolute difference in fold change. The top few genes increasing and decreasing in expression are shown in *Figure 2E*.

## Single-cell gene expression data collection and analyses

For single-cell gene expression analyses, human forebrain assembloids (hFA) were infected with Lenti Dlx1/2b::eGFP as described above. Samples were incubated either under a normoxic environment (37 °C, 5% $CO_2$, 21% $O_2$), or under the controlled hypoxic environment of the HypoxyLab (Oxford Optronix). Following the environmental exposure, hFA were dissected under a microscope to separate the hFA into hSO and hCO. To analyze only interneurons migrated within the hCO and not the ones still at the edges of the fusion, we intentionally performed the separation of hFA more toward the hCO side. For single-cell gene expression analysis, organoids were dissociated as previously described (*Valenzuela-Sánchez et al., 2016*; *Marín, 2016*). Dissociated cells were resuspended in ice-cold PBS containing 0.02% BSA and loaded onto a Chromium Single cell 3' chip (with an estimated recovery of 10,000 cells per channel) to generate gel beads in emulsion (GEMs). scRNA-seq libraries were prepared with the Chromium Single cell 3' GEM, Library & Gel Bead Kit v2 (10 x Genomics, PN: 120237). Libraries from different samples were pooled and sequenced on a NovaSeq S4 (Illumina) using 150×2 chemistry. Sample demultiplexing, barcode processing, and unique molecular identifiers counting were performed using the Cell Ranger software suite (v6.1.2 with default settings excluding introns). Reads were then aligned to the human reference genome (GRCh38), filtered, and counted. Expression matrices for the samples were processed using the R (v4.1.2) package Seurat (v4.2). We excluded cells that express less than 2500 genes as well as those with a proportion of mitochondrial reads higher than 15%. Gene expression was then normalized using a global-scaling normalization method (normalization.method='LogNormalize', scale.factor=10,000), and the 2000 most variable genes were then selected (selection.method='versust'). The *Harmony* package (*Marín, 2012*) was used to integrate control and hypoxia samples for downstream analysis. The top 7 and 6 principal components for hCO and hSO samples, respectively, were utilized to perform a UMAP projection, implemented with the *RunUMAP* function with default settings. The clusters were generated with resolutions of 0.1 and 0.3, respectively, for hCO and hSO samples using the *FindNeighbors* and *FindClusters* functions. We identified clusters based on differential expression of known markers (*Kepecs and Fishell, 2014*) identified using the FindAllMarkers function with default settings. In some cases, using expressions of known markers, we grouped together clusters that were originally separate based on the Seurat clustering. For visualization purposes, the color palette of UMAP plots for hSO and hCO was adjusted using 'magic wand' in non-contiguous mode for selection of all cells from respective clusters, for subsequent synchronization of the color for the given cell cluster between brain regions (Adobe Photoshop CS 2018). Original plots are available upon request. Markers used to verify the

hypoxic response were based on *PDK1* and *PFKP*. Proportions of cells expressing *ADM* (Expression Level >0) out of all cells per cluster were calculated using (*Fukuchi et al., 2014*).

## Cryopreservation and cryosection of hSO

Whole hSOs that were infected with Dlxi1/2b::eGFP⁺ were fixed with 4% paraformaldehyde overnight. They were then washed with PBS and transferred to 30% sucrose for up to 72 hr. The samples were transferred into an embedding medium block (Tissue-Tek OCT Compound 4583, Sakura Finetek), snap frozen and sectioned at 10-µm-thick microns using a cryostat (Leica).

## Immunocytochemistry and quantification of cleaved-caspase 3

The cryosections of hSO were washed with PBS three times for 5 min each to remove excess OCT and then blocked for 1 hr in PBS +0.5% triton X-100+5% BSA+0.5% DMSO at room temperature. The slides were then incubated with cleaved-caspase 3 (anti-rabbit, 1:100, Cell Signaling Technology, 9661T) overnight at 4 °C. After incubation, the slides were washed three times with PBS +0.5% triton X-100 for 5 min each. Secondary antibody incubation was then performed with Alexa Fluor 594 (anti-rabbit, 1:1000, Invitrogen, Cat # A-21207), anti-GFP (1:1000, GeneTex, GTX13970) for 1 hr at room temperature. After, slides were washed two times with PBS for 5 min each and then stained with Hoescht 33258 (1:10,000, Life Technologies) for 5 min at room temperature. Cryosections were mounted on glass coverslips using aquamount (Thermo Fisher Scientific) and were imaged using (Zeiss Axio Observer). Images were visualized and the number of cleaved-caspase 3 (c-CAS3) and Dlxi1/2b::eGFP⁺ positive cells were quantified using Fiji ImageJ Version 1.54 p.

## Flow cytometry analyses for Annexin V

To assess the levels of apoptotic Dlxi1/2b::mScarlet cells, FACS was conducted on NovoCyte Penteon instrument using NovoExpress 1.5.6 software (Stanford FACS Facility). To label inhibitory interneurons in hSOs at later stages of development, hiPSC lines were engineered to express mScarlet under the dlx1/2 promoter. Briefly, LV-Dlxi1/2b-mScarlet lentiviral particles were produced according to the protocol described in this manuscript. Control iPSC lines were plated sparsely and infected with the lentivirus at 1:300 dilution in E8 medium for 24 hr, following which the media was changed to E8 medium without lentivirus. The cells were dissociated using Accutase (Innovate Cell Technologies, AT-104) and plated sparsely in six-well plates for Puromycin (Sigma-Aldrich, P8833) selection of single cell clones with successful genomic integration of the LV-Dlxi1/2b-mScarlet construct. Selected clones were used to make hSO organoids according to the previously described protocol and used for further experiments. On the day of analysis, 3 spheres per line were placed in an Eppendorf tube and dissociated in 300 µL of Accutase at 37 °C for 30 min. After incubation, 1200 µL of neurobasal medium was added to each tube to inactivate the Accutase. The cells were then centrifuged at 200 × *g* for 4 min and resuspended in DPBS (Thermo Fisher Scientific, 14190144) and strained using a 40 µM cell strainer (Biologix Research Company, 15–1040). Using the Annexin V kit (Invitrogen, V13246), the cells were stained with Annexin V, Alexa Fluor-647 (Invitrogen, A23204) in RT for 15 min. The analysis of Annexin V signals in Dlxi1/2b::mScarlet cells was done using FlowJo (v.10.8.1).

## Real-time quantitative PCR (qPCR)

mRNA was isolated using the RNeasy Mini Kit and RNase-Free DNase set (Qiagen, 74136), and template cDNA was prepared by reverse transcription using the PowerUp SYBR Green master mix for qRT-PCR (Life Technologies, A25742). Real-time qPCR was performed on a CFX384 Real-time system (Bio-Rad). Data was processed using the Bio-Rad CFX Maestro (Bio-Rad). Primers used are listed in *Supplementary file 2*.

## Western blot assays

Whole hSOs were rapidly lysed by gentle agitation on ice using the RIPA buffer system with protease and a phosphatase inhibitor cocktail (Santacruz, sc-24948A) for HIF1α, RAMP1, and RAMP2 analyses. Whole cell lysates were incubated at 4 °C for 1 hr, centrifuged at 14,000 x *g* for 15 min. After the supernatant was collected, and the protein concentration of the supernatant was measured using the Pierce BCA Protein Assay Kits (Thermo Fisher Scientific, 23225). The lysate was denatured in Bolt LDS Sample Buffer (Invitrogen, B0007) at 95 °C for 5 min. The samples (at 7 µg) were loaded on a Bolt

4–12% Bis-Tris Protein Gels (Invitrogen, NW04120BOX) and then transferred to a PVDF membrane using iBlot 2 Transfer Stacks (Invitrogen, IB24002) and an iBlot2 (Method: P3; Thermo Fisher Scientific, IB24002). After the samples were blocked with 5% skim milk (BD, 232100) in TBST (Tris-buffered saline [Boston BioProducts, BM-301X] containing 0.1% Tween 20 [Sigma-Aldrich, P1379 TBST]) for 1 hr, they were incubated in the blocking solution (TBST containing 5% BSA [Gendepot, A0100-005]) with the following primary antibodies: β-actin (anti-mouse, Clone 13E5, 4970 S), HIF1α (anti-rabbit, Clone D2U3T, 14179 S), RAMP1 (rabbit, 1:500, Abcam, ab156575), RAMP2 (mouse, 1:500, Santacruz, sc-365240), α-tubulin (anti-rat, 1:1000; Abcam, ab6160) at 4 °C overnight. After the blot was washed with 5% TBST, the membranes were incubated in the blocking solution with the following horseradish peroxidase-conjugated secondary antibodies: Anti-rat IgG (1:2000; Cell Signaling, #7077), Anti-mouse IgG (1:2000; Cell Signaling, #7076), Anti-rabbit IgG (1:2000; Cell Signaling, #7074) at RT for 1 hr. The SuperSignal West Femto Maximum Sensitivity Substrate (Thermo Fisher Scientific, 34095) was used to develop the membranes. The iBright 1500 (Thermo Fisher Scientific, A44114) was used to detect and analyze protein bands. The intensity of each band was quantified with ImageJ (1.53t) software (NIMH, Bethesda, MD) with normalization to background β-actin or α-tubulin.

## Enzyme immunoassay

ADM levels were quantified using the competitive enzyme immunoassay for human ADM (Phoenix Pharmaceuticals, EK-010–01) and assays were performed according to the manufacturer's recommendation. Calculations of sample concentration were performed based on peptide standard solution in the range of 0–100 ng/mL. The concentrations of the samples were $log_{10}$ transformed, and concentrations of the samples were interpolated from the sigmoidal curve. The range of reliable detection was between 0.01100 ng/mL. Samples with concentration <0.01 ng/mL were quantified as 0.

The quantification of the pAKT:AKT(pS473) ratio and pERK:ERK (pT202/pY204, pT185/pY187) ratio was done using the AKT (Total/Phospho) InstantOne ELISA Kit (Thermo Fisher Scientific, 85-86046-11) for pAKT:AKT, and the ERK1/2 (Total/Phospho) InstantOne ELISA Kit (Thermo Fisher Scientific, 8586013–11) for pERK:ERK. Both assays were performed according to manufacturers' protocol. For both assays, samples were lysed in a 200 µL lysis buffer provided by the manufacturer. Samples were incubated for 1 hr, and 50 µL of lysate per well was used each of the assays. Quantification of cAMP levels was performed using the cAMP Assay Kit (Competitive ELISA) (Abcam, ab65355) according to manufacturers' protocol. For the analysis, samples were lysed in 250 µL lysis buffer, and 100 µL was used for the assay. The concentration of cAMP was calculated based on the standard curve obtained from kit standards. The quantification of PKA was done using the PKA (Protein Kinase A) Colorimetric Activity Kit (Thermo Fisher Scientific, EIA PKA) according to manufacturers' protocol. For the assay, samples were lysed in 200 µL lysis buffer containing protease inhibitor cocktail (1 µL/mL, Sigma-Aldrich, P1860), phenylmethylsulfonyl fluoride (1 mM, Sigma-Aldrich, 10837091001), and Activated Sodium Orthovanadate (10 mM, Calibiochem, 567540). Samples were incubated for 1 hr under shaking (250 rpm) and centrifuged at 10,000 rpm for 10 min at 4 °C. Samples were then diluted 1 x with kit dilution buffer, and 40 µL of diluted sample was used for analysis. The quantification of the pCREB:CREB ratio was done using the phosphoELISA CREB (pS133) (Thermo Scientific, KHO0241) and phosphoELISA CREB (Total; Thermo Scientific, KHO0231) according to manufacturer's protocol. For both assays, samples were lysed in 100 µl RIPA lysis buffer containing phosphatase inhibitor cocktail (Santa Cruz Biotechnology, sc-45065) and a 1:50 dilution of the lysates in standard diluent buffer was used for the assay. The concentration of pCREB and CREB (total) was calculated based on the standard curve obtained from the kit standards using the Tecan Infinite M1000 Pro and i-control 1.10 software.

## Exogenous ADM peptide denaturation and LC-MS analysis

The synthetic peptide ADM (2 nmol, Anaspec, AS-60447) was incubated with either $ddH_20$ (for control condition) or Dithiothreitol (DTT; 5 mM, Sigma-Aldrich, D5545), at 56 °C for 30 min. Following reduction of disulfide bridges, alkylation of the free SH-groups was performed with iodoacetamide (IAM; 20 mM, Sigma-Aldrich, I1149) at 23 °C for 45 min in the dark. The unreacted IAM was quenched with 5 mM final DTT. The solution was desalted and concentrated using Amicon Ultra (3kD, EMD Millipore, UFC500324), and the volume was adjusted to the starting volume with UltraPure DNase/RNase Free Distilled Water (Thermo Fisher Scientific, 10-977-015). LC-MS analysis was performed

using Vanquish LC system coupled with an ID-X Tribrid mass spectrometry equipped with a heated electrospray ionization source (HESI-II, Thermo Fisher Scientific). For the LC system, a ZORBAX RRHD diphenyl 100x2.1 mm column (Agilent, 858750–944) was used. The mobile phases were 0.1% formic acid (Fisher Scientific, A117-50) in ddH$_2$O for A and 0.1% formic acid in acetonitrile (Sigma-Aldrich, 34998) for B. The chromatographic gradient was as follows: 0–2 min 10% B, linear increase 2–10 min up to 60% B, linear increase 10–15 min up to 98% B, hold 15–17 min at 98% B, linear decrease 17–18 min to 10% B, equilibrate 18–21 min at 10% B. The mobile phase flow rate was set to 250 µL/min with an injection volume of 3 µL. The autosampler temperature was set to 4 °C, and the column chamber temperature to ambient. Data acquisition on the IDX MS was acquired at 120,000 resolution setting with positive ion voltage of 3.5 kV, vaporizer and transfer tube temperature of 325 °C; AGC target set to standard, RF lens set to 45%, sheath gas set to 40 AU, aux gas set to 12 AU, maximum injection time of 100ms, with 5 microscans/acquisition, and scan range of 500–2000 *m/z*. Examination of the LC-MS data was performed using Xcalibur FreeStyle 1.6 (Thermo Fisher Scientific). Here, we monitored the formation of carbamidomethyl-cysteines at Cys16-Cys21 (predicted M+1: 6142.9973), or the reduced form of the peptide (predicted M+1: 6028.9544).

## Statistical analyses

Data are presented as mean ± s.e.m. unless otherwise indicated. Distribution of the raw data was tested for normality; statistical analyses were performed as indicated in figure legends. Sample sizes were estimated empirically or based on power calculations. Blinding was used for analyses comparing control and treatment samples.

## Acknowledgements

Stanford Maternal & Child Health Research Institute (MCHRI) Postdoctoral Fellowship (to LL), Knut and Alice Wallenberg Foundation Postdoctoral Fellowship (to WPM), Swedish Research Council IPD (to WPM), Bill and Melinda Gates Foundation (to AMP).

## Additional information

### Competing interests

Fikri Birey: listed as inventor for U.S. Patent number 10,676,715. The author has no other competing interests to declare. Anca M Pasca: is listed as inventor for U.S Patent number: 10,494,602 B2. The author has no other competing interests to declare. The other authors declare that no competing interests exist.

### Funding

| Funder | Grant reference number | Author |
| --- | --- | --- |
| Knut and Alice Wallenberg Foundation | | Wojciech P Michno |
| Stanford Maternal and Child Health Research Institute | | Li Li |
| Swedish Research Council | | Wojciech P Michno |
| Gates Foundation | | Anca M Pasca |

The funders had no role in study design, data collection and interpretation, or the decision to submit the work for publication.

### Author contributions

Alyssa Puno, Conceptualization, Data curation, Formal analysis, Validation, Investigation, Visualization, Methodology, Writing – original draft, Writing – review and editing; Wojciech P Michno, Conceptualization, Data curation, Formal analysis, Supervision, Validation, Investigation, Visualization, Methodology, Writing – original draft, Project administration, Writing – review and editing; Li Li, Conceptualization,

Data curation, Software, Formal analysis, Visualization, Methodology, Project administration, Writing – review and editing; Amanda Everitt, Data curation, Software, Formal analysis, Visualization, Methodology, Writing – original draft, Writing – review and editing; Kate McCluskey, Data curation, Software, Formal analysis, Investigation, Visualization, Methodology, Writing – original draft, Writing – review and editing; Saw Htun, Seyeon Park, Data curation, Formal analysis, Validation, Methodology, Writing – review and editing; Dhriti Nagar, Data curation, Formal analysis, Supervision, Investigation, Visualization, Methodology, Writing – review and editing; Jong Bin Choi, Formal analysis, Supervision, Validation, Investigation, Visualization, Methodology, Writing – review and editing; Yuqin Dai, Formal analysis, Supervision, Investigation, Writing – review and editing; Emily Gurwitz, Data curation, Formal analysis, Validation, Writing – review and editing; Jeremy A Willsey, Data curation, Software, Formal analysis, Supervision, Investigation, Visualization, Methodology, Writing – review and editing; Fikri Birey, Data curation, Formal analysis, Supervision, Visualization, Methodology, Writing – review and editing; Anca M Pasca, Conceptualization, Resources, Data curation, Formal analysis, Supervision, Funding acquisition, Validation, Investigation, Visualization, Methodology, Writing – original draft, Project administration, Writing – review and editing

**Author ORCIDs**
Alyssa Puno ⬡ https://orcid.org/0009-0001-8342-5324
Wojciech P Michno ⬡ https://orcid.org/0000-0002-3096-3604
Jong Bin Choi ⬡ https://orcid.org/0000-0002-7017-1775
Anca M Pasca ⬡ https://orcid.org/0000-0002-0445-9009

Reviewer #1 (Public review): https://doi.org/10.7554/eLife.108134.3.sa1
Reviewer #2 (Public review): https://doi.org/10.7554/eLife.108134.3.sa2
Reviewer #3 (Public review): https://doi.org/10.7554/eLife.108134.3.sa3
Author response https://doi.org/10.7554/eLife.108134.3.sa4

## Additional files

### Supplementary files
Supplementary file 1. List of hiPSC lines and number of cells used for each experiment.

Supplementary file 2. List of primers used for qPCRs.

Supplementary file 3. List of differentially expressed genes by RNA-Sequencing in hSOs exposed to hypoxia.

Supplementary file 4. List of GSEA terms from RNA-Sequencing.

MDAR checklist

### Data availability
Gene expression data are available in the Gene Expression Omnibus (GEO) under accession number GSE183201.

The following dataset was generated:

| Author(s) | Year | Dataset title | Dataset URL | Database and Identifier |
|---|---|---|---|---|
| Pasca AM, Willsey JA | 2023 | Adrenomedullin promotes the migration of interneurons in a human forebrain assembloid model for hypoxic interneuronopathy of prematurity | https://www.ncbi.nlm.nih.gov/geo/query/acc.cgi?acc=GSE183201 | NCBI Gene Expression Omnibus, GSE183201 |

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

## Appendix 1

**Appendix 1—key resources table**

| Reagent type (species) or resource | Designation | Source or reference | Identifiers | Additional information |
|---|---|---|---|---|
| Cell line (*Homo sapiens*) | hiPSC lines derived from fibroblasts | *Paşca et al., 2015*; *Paşca et al., 2019* | No other identifiers available or required for publication based on previous publications with these lines | |
| Transfected construct (Dlx) | Dlxi1/2b-mScarlet lentiviral | *Birey et al., 2022* | 254611 | Lentiviral construct to transfect and express the shRNA. |
| Antibody | β-actin (anti-rabbit) Clone 13E5 Rabbit Monoclonal | Cell Signaling Technology | 4970 S RRID:AB_3740851 | WB (1:1000) |
| Antibody | HIF1α (anti-rabbit Clone D2U3T, Rabbit Monoclonal) | Cell Signaling Technology | 14179 S RRID:AB_2622225 | WB (1:1000) |
| Antibody | RAMP1 (anti-rabbit) Rabbit Recombinant Monoclonal | Abcam ab156575 | ab156575 RRID:AB_2801501 | WB (1:500) |
| Antibody | RAMP2 (anti-mouse) Mouse Monoclonal | Santacruz | sc-365240 RRID:AB_10844326 | WB (1:500) |
| Antibody | α-tubulin (anti-rat) Rat Monoclonal | Abcam | ab6160 RRID:AB_305328 | WB (1:1000) |
| Antibody | Anti-rat IgG Polyclonal | Cell Signaling Technology | #7077 RRID:AB_10694715 | WB (1:2000) |
| Antibody | Anti-mouse IgG, HRP-linked | Cell Signaling | #7076 RRID:AB_330924 | WB (1:2000) |
| Antibody | Anti-rabbit IgG | Cell Signaling | #7074 RRID:AB_2099233 | WB (1:2000) |
| Antibody | Annexin V-Alexa Fluor-647 | Invitrogen | A23204 RRID:AB_3740829 | IF (1:1000) |
| Antibody | Alexa Fluor 594 (anti-rabbit) | Invitrogen | A-21207 RRID:AB_141637 | IF (1:1000) |
| Antibody | anti-GFP Chicken Polyclonal | GeneTex | GTX13970 RRID:AB_371416 | IF (1:1000) |
| Antibody | Hoescht 33258 | Life Technologies | 33258 | IF (1:10,000) |
| Antibody | cleaved-caspase 3 (anti-rabbit) Rabbit Polyclonal | Cell Signaling Technology | 9661T RRID:AB_3740831 | IF (1:100) |
| Commercial assay or kit | Chromium Single cell 3' GEM, Library & Gel Bead Kit v2 | 10 x Genomics | PN: 120237 | |
| Commercial assay or kit | Annexin V kit | Invitrogen | V13246 | FACS (1:20) |
| Commercial assay or kit | The SuperSignal West Femto Maximum Sensitivity Substrate | Thermo Fisher Scientific | 34095 | |
| Commercial assay or kit | RNeasy Mini Kit and RNase-Free DNase set | Qiagen | 74136 | |

*Appendix 1 Continued on next page*

*Appendix 1 Continued*

| Reagent type (species) or resource | Designation | Source or reference | Identifiers | Additional information |
|---|---|---|---|---|
| Commercial assay or kit | PowerUp SYBR Green master mix | Life Technologies | A25742 | |
| Commercial assay or kit | Pierce BCA Protein Assay Kits | Thermo Fisher Scientific | 23225 | |
| Commercial assay or kit | Bolt LDS Sample Buffer | Invitrogen | B0007 | |
| Commercial assay or kit | iBlot 2 Transfer Stacks | Invitrogen | IB24002 | |
| Commercial assay or kit | iBlot2 | Thermo Fisher Scientific | IB24002 | |
| Commercial assay or kit | 5% skim milk | BD | 232100 | |
| Commercial assay or kit | TBST (Tris-buffered saline) | Boston BioProducts | BM-301X | |
| Commercial assay or kit | 0.1% Tween 20 | Sigma-Aldrich | P1379 | |
| Commercial assay or kit | blocking solution (TBST) containing 5% BSA | Gendepot | A0100-005 | |
| Commercial assay or kit | Adrenomedullin (ADM) competitive enzyme immunoassay | Phoenix Pharmaceuticals | EK-010–01 | |
| Commercial assay or kit | pAKT:AKT(pS473) AKT (Total/Phospho) InstantOne ELISA Kit | Thermo Fisher Scientific | 85-86046-11 | |
| Commercial assay or kit | ERK1/2 (Total/Phospho) InstantOne ELISA Kit | Thermo Fisher Scientific | 8586013–11 | |
| Commercial assay or kit | cAMP Assay Kit (Competitive ELISA) | Abcam | ab65355 | |
| Commercial assay or kit | PKA (Protein Kinase A) Colorimetric Activity Kit | Thermo Fisher Scientific | EIA PKA | |
| Commercial assay or kit | phosphoELISA CREB (pS133) | Thermo Scientific | KHO0241 | |
| Commercial assay or kit | phosphoELISA CREB (Total) | Thermo Scientific | KHO0231 | |
| Chemical compound, drug | ROCK inhibitor Y-27632 | Selleck Chemicals | S1049 | |
| Chemical compound, drug | dorsomorphin | SigmaAldrich | P5499 | |
| Chemical compound, drug | SB-431542 | Tocris | 1614 | |
| Chemical compound, drug | XAV-939 | Tocris | 3748 | |
| Chemical compound, drug | SHH pathway agonist SAG | Thermo Fisher Scientific | 566660 | |

*Appendix 1 Continued on next page*

*Appendix 1 Continued*

| Reagent type (species) or resource | Designation | Source or reference | Identifiers | Additional information |
|---|---|---|---|---|
| Chemical compound, drug | EGF | R&D Systems | 236-EG | |
| Chemical compound, drug | FGF2 | R&D Systems | 233-FB | |
| Chemical compound, drug | BDNF | Peprotech | 50–02 | |
| Chemical compound, drug | NT3 | Peprotech | 450–03 | |
| Chemical compound, drug | Puromycin | Sigma-Aldrich | P8833 | |
| Chemical compound, drug | RIPA lysis buffer containing phosphatase inhibitor cocktail | Santa Cruz Biotechnology | sc-45065 | |
| Chemical compound, drug | protease inhibitor cocktail | Sigma-Aldrich | P1860 | |
| Chemical compound, drug | phenylmethylsulfonyl fluoride | Sigma-Aldrich, | 10837091001 | |
| Chemical compound, drug | Activated Sodium Orthovanadate | Calibiochem, | 567540 | |
| Chemical compound, drug | Adrenomedullin (ADM) | Anaspec | AS-60447 | |
| Chemical compound, drug | $ADM_{22-52}$ (the RAMP2 receptor blocker) | Cayman Chemical Company | 24892 | |
| Chemical compound, drug | Dithiothreitol (DTT) | Sigma Aldrich | D5545 | |
| Chemical compound, drug | Iodoacetamide (IAM) | Sigma-Aldrich | I1149 | |
| Chemical compound, drug | 0.1% formic acid | Fisher Scientific | A117-50 | |
| Chemical compound, drug | formic acid in acetonitrile | Sigma-Aldrich | 34998 | |
| Software, algorithm | Tecan Infinite M1000 Pro and i-control 1.10 | Tecan | RRID:SCR_025732 | |
| Software, algorithm | Xcalibur FreeStyle 1.6 | Thermo Scientific | RRID:SCR_014593 | |
| Software, algorithm | ImageJ (1.53t) software | NIH | RRID:SCR_003070 | |

*Appendix 1 Continued on next page*

*Appendix 1 Continued*

| Reagent type (species) or resource | Designation | Source or reference | Identifiers | Additional information |
|---|---|---|---|---|
| Software, algorithm | BioRad CFX Maestro | BioRad | RRID:SCR_018064 | |
| Software, algorithm | NovoExpress 1.5.6 software | NovoCyte Penteon | RRID:SCR_024676 | |
| Software, algorithm | Cell Ranger software suite v6.1.2 | 10 X Genomics | RRID:SCR_017344 | |
| Software, algorithm | R (v4.1.2) package Seurat (v4.2) | *Hao et al., 2021* | RRID:SCR_001905 RRID:SCR_007322 | RNA-seq analysis |
| Software, algorithm | STAR v2.7.3 | *Dobin et al., 2013* | RRID:SCR_004463 | GRCh38 human genome reference |
| Software, algorithm | Picard (v 2.21.1) | https://github.com/broadinstitute/picard | RRID:SCR_006525 | Alignment and RNA-Seq quality control |

