## [Editor Report · eLife Assessment]

The authors combined human assembloids, fetal brain tissue, bulk and single cell RNA sequencing, and live imaging to understand the molecular mechanisms affected by hypoxia during cortical development. The findings are very **important** to the neurodevelopmental field, They reveal new insights into how migration of cortical interneurons can be affected in hypoxic conditions, and provide exciting models to probe broad neurodevelopmental processes in health and disease. The evidence is **compelling**. The data and analyses are very rigorous and go beyond the state-of-the-art.

---

## [Referee Report · Reviewer #1 (Public review)]

Summary:

This work aims to elucidate the molecular mechanisms affected in hypoxic conditions causing reduced cortical interneuron migration. They use human assembloids as a migratory assay of subpallial interneurons into cortical organoids and show substantially reduced migration upon 24 hours hypoxia. Bulk and scRNA-seq shows adrenomedullin (ADM) up-regulation, as well as its receptor RAMP2 confirmed at protein level. Adding ADM to the culture medium after hypoxic conditions rescues the migration deficits, even though the subtype of interneurons affected is not examined. However, the authors demonstrate very clearly that ineffective ADM does not rescue the phenotype and blocking RAMP2 also interferes with the rescue. The authors are also applauded for using 4 different cell lines and using human fetal cortex slices as an independent method to explore the DLXi1/2GFP-labelled iPSC-derived interneuron migration in this substrate with and without ADM addition (after confirming that also in this system ADM is up-regulated). Finally, the authors demonstrate PKA - CREB signalling mediating the effect of ADM addition, and also lead to up-regulation of GABAreceptors. Taken together this is a very carefully done study on an important subject - how hypoxia affects cortical interneuron migration. In my view it would be of great interest for the readers of Elife.

Strengths:

Its strengths are the novelty and the thorough work using several culture methods and 4 independent lines.

Weaknesses:

The main weakness is that we dont know which interneuron subtypes are most affected by hypoxia and which may be rescued in their migration by ADM.

A further weakness is that the few genes confirmed to be regulated after hypoxia do not help determining which statistical cut-off can be considered reliable, given that they didn't compare strongly regulated versus weakly regulated genes.

Comments on revisions:

Unfortunately, the authors did not address my suggestions. While they show example stainings of interneuron subtypes, they do not show if Calretinin, calbinin or somatostatin+ interneurons are differentially affected by hypoxia or the rescue with ADM. I still consider this an important piece of information to add.

---

## [Referee Report · Reviewer #2 (Public review)]

Summary:

The manuscript by Puno and colleagues investigates the impact of hypoxia on cortical interneuron migration and downstream signaling pathways. They establish two models to test hypoxia, cortical forebrain assembloids and primary human fetal brain tissue. Both of these models provide a robust assay for interneuron migration. In addition, they find that ADM signaling mediates the migration deficits and rescue using exogenous ADM. The findings are novel and very interesting to the neurodevelopmental field, revealing new insights into how cortical interneurons migrate and as well, establishing exciting models for future studies.The authors use sufficient iPSC lines including both XX and XY, so analysis is robust. In addition, the RNAseq data with re-oxygenation is a nice control to see what genes are changed specifically due to hypoxia. Further, the overall level of valiation of the sequencing data and involvement of ADM signaling is convincing, including the validation of ADM at the protein level. Overall this is a very nice manuscript. I have a few comments and suggestions for the authors.

Strengths/Weaknesses:

(1) Can they comment on the possibility of inflammatory response pathways being activated by hypoxia - has this been shown before? While not the focus of the manuscript, it would be discussed in the Discussion as an interesting finding and potential involvement of other cells in the Hypoxic response.

(2) Can they comment on the mechanism at play here with respect to ADM and binding to RAMP2 receptors - is this a potential autocrine loop, or is the source of ADM from other cell types besides inhibitory neurons? Given the scRNA-seq data, what cell-to-cell mechanisms can be at play? Since different cells express ADM, there could be different mechanisms at place in ventral vs dorsal areas.

(3) For data from Figure 6 - while the ELISA assays are informative to determine which pathways (PKA, AKT, ERK) are active, there is no positive control to indicate these assays are "working" - therefore, if possible, western blot analysis from assembloid tissue could be used (perhaps using the same lysates from Fig 3) as an alternative to validate changes at the protein level (however, this might prove difficult); further to this, is P-CREB activated at the protein level using WB?

(4) Can the authors comment further on the mechanism and what biological pathways and potential events are downstream of ADM binding to RAMP2 in inhibitory neurons? What functional impact would this have linked to the CREB pathway proposed? While the link to GABA receptors is proposed, CREB has many targets beyond this.

(5) Does hypoxia cause any changes to inhibitory neurogenesis (earlier stages than migration?) - this might always be known but was not discussed.

(6) In the Discussion section - it might be worth detailing to the readers what the functional impact of delayed/reduced migration of inhibitory neurons into the cortex might results in, in terms of functional consequences for neural circuit development

Comments on revisions:

The authors have addressed my comments thoroughly. I have no further comments or suggestions

---

## [Referee Report · Reviewer #3 (Public review)]

Summary:

The authors aimed to test whether hypoxia disrupts the migration of human cortical interneurons, a process long suspected to underlie brain injury in preterm infants but previously inaccessible for direct study. Using human forebrain assembloids and ex vivo developing brain tissue, they visualized and quantified interneuron migration under hypoxic conditions, identified molecular components of the response, and explored the effect of pharmacological intervention (specifically ADM) on restoring the migration deficits.

Strengths:

The major strength of this study lies in its use of human forebrain assembloids and ex vivo prenatal brain tissue, which provide a direct system to study interneuron migration under hypoxic conditions. The authors combine multiple approaches: long-term live imaging to directly visualize interneuron migration, bulk and single-cell transcriptomics to identify hypoxia-induced molecular responses, pharmacological rescue experiments with ADM to establish therapeutic potential, and mechanistic assays implicating the cAMP/PKA/pCREB pathway and GABA receptor expression in mediating the effect. Together, this rigorous and multifaceted strategy convincingly demonstrates that hypoxia disrupts interneuron migration and that ADM can restore this defect through defined molecular mechanisms.

Overall, the authors achieve their stated aims, and the results strongly support their conclusions. The work has significant impact by providing the first direct evidence of hypoxia-induced interneuron migration deficits in the human context, while also nominating a candidate therapeutic avenue. Beyond the specific findings, the methodological platform-particularly the combination of assembloids and live imaging-will be broadly useful to the community for probing neurodevelopmental processes in health and disease.

Comments on revisions:

The authors have fully addressed my concerns by incorporating the relevant discussion into the manuscript, especially regarding how well the migration observed in hSO-hCO assembloids reflects in vivo condition. I have no further comments.

---

## [Author Response]

The following is the authors’ response to the original reviews.

**Public Reviews:**

**Reviewer #2 (Public review):**
Weaknesses:(1) Can the authors comment on the possibility of inflammatory response pathways being activated by hypoxia? Has this been shown before? While not the focus of the manuscript, it could be discussed in the Discussion as an interesting finding and potential involvement of other cells in the Hypoxic response.

We thank the reviewer for reviewing our manuscript and for the important comment about inflammation. Indeed, hypoxia has been shown to activate the inflammatory response pathways. In various studies, it was found that HIF-1a can interact with NF-κB signaling, leading to the upregulation of pro-inflammatory cytokines such as IL-1β, IL-6, and TNF-α (Rius et al., Cell, 2008; Hagberg et al., Nat Rev Neurol, 2015).

In our transcriptomics data (Fig. 2D), and to the reviewers’ point, we identified enrichment of inflammatory signaling response following the hypoxic exposure. Since hSO at the time of analyses do contain some astrocytes, we think these contribute to the observed pro-inflammatory changes and emphasize the feasibility of capturing this response in organoids in vitro. This is also important because ADM is known to have anti-inflammatory properties and should be investigated as such in future studies focused on hypoxia-induced inflammation.

In the manuscript, we included a few sentences in the discussion to address the lack of in-depth analyses of inflammation as a limitation of our study.

(2) Could the authors comment on the mechanism at play here with respect to ADM and binding to RAMP2 receptors - is this a potential autocrine loop, or is the source of ADM from other cell types besides inhibitory neurons? Given the scRNA-seq data, what cell-to-cell mechanisms can be at play? Since different cells express ADM, there could be different mechanisms in place in ventral vs dorsal areas.

Based on our scRNA-seq data in hSOs showing significant upregulation of ADM expression in astrocytes and progenitors, and increased expression of RAMP2 receptors on neurons, we speculate that the primary mechanism is likely to involve paracrine interactions. However, we cannot exclude autocrine mechanisms with the current experiments. Dissecting these interactions in a cell-type specific manner could be an important focus for future ADM-related studies.

To address the question about the possible different mechanisms in ventral versus dorsal areas, in the revision, we plotted and included in the figures the data about the cell-type expression of ADM and its receptors in hCOs (Fig. S3)

(3) For data from Figure 6 - while the ELISA assays are informative to determine which pathways (PKA, AKT, ERK) are active, there is no positive control to indicate these assays are "working" - therefore, if possible, western blot analysis from assembloid tissue could be used (perhaps using the same lysates from Figure 3) as an alternative to validate changes at the protein level (however, this might prove difficult); further to this, is P-CREB activated at the protein level using WB?

We thank the reviewer for this comment and the observation. Although we did not include a traditional positive control in these ELISA assays, several lines of evidence indicate that the measurements are reliable. First, the standard curves behaved as expected, and all sample values fell within the assay’s dynamic range. Second, technical replicates showed low variability, and the observed changes across experimental conditions (e.g., hypoxia vs. control) were consistent with the expected biological responses based on previous literature. We agree that including western blot validation would strengthen the findings, and we will note this for our future studies focused on CREB and ADM.

(4) Could the authors comment further on the mechanism and what biological pathways and potential events are downstream of ADM binding to RAMP2 in inhibitory neurons? What functional impact would this have linked to the CREB pathway proposed? While the link to GABA receptors is proposed, CREB has many targets beyond this.

We appreciate the reviewers’ insightful question. Currently, not much is known about the molecular pathways and downstream cellular events triggered by ADM binding to RAMP2 in inhibitory neurons, and in general in brain cells. The data from our study brings the first information about the cell-type specific expression of ADM in baseline and hypoxic conditions and is one of the key novelties of our study.

While the signaling landscape of ADM in interneurons is largely unexplored, several studies in other (non-brain) cell types have demonstrated that ADM binding to RAMP2 can activate downstream cascades such as the cAMP/PKA/CREB pathway, PI3K/AKT, and ERK/MAPK, all of which are also known to be critical regulators of neuronal development and survival. These previously published data along with our CREB-targeted findings in hypoxic interneurons, suggest ADM–RAMP2 signaling could influence multiple aspects of interneuron biology, but these remain to be evaluated in future studies.

We agree with the reviewer that CREB has a wide range of transcriptional targets. We decided to focus on GABA as a target of CREB for two main reasons, including: (i) GABA signaling has been previously shown to play an important role in the migration of cortical interneurons, and (ii) a previous study by Birey et al. (Cell Stem Cell, 2022) demonstrated that CREB pathway activity is essential for regulating interneuron migration in assembloid models of Timothy Syndrome, thus further providing evidence that dysregulation of CREB activity disrupts migration dynamics.

While our study provides a first step toward uncovering the mechanisms of interneuron migration protection by ADM, we fully acknowledge that future work will be needed to delineate the full spectrum of ADM–RAMP2 downstream signaling events in inhibitory neurons and other brain cells.

(5) Does hypoxia cause any changes to inhibitory neurogenesis (earlier stages than migration?) - this might always be known, but was not discussed.

We appreciate this question from the reviewer; however, this was not something that we focused on in this manuscript due to the already large amount of data included. A separate study focusing on neurogenesis defects and the molecular mechanisms of injury for that specific developmental process would be an important next step.

(6) In the Discussion section, it might be worth detailing to the readers what the functional impact of delayed/reduced migration of inhibitory neurons into the cortex might result in, in terms of functional consequences for neural circuit development.

We thank the Reviewer for the suggestion of detailing the functional impact of reduced inhibitory neuron migration. The manuscript to discuss that previous studies show that failure of interneurons to migrate and reach their designated targets within the appropriate developmental window leads to their elimination through apoptosis. Decreased numbers (or abnormal development) of interneurons are associated with neurodevelopmental impairments and abnormal functional connectivity in the brain.

**Recommendations for the authors:**

**Reviewer #1 (Recommendations for the authors):**
(1) The authors should examine if all cortical interneurons are affected by ADM or only subtypes (Parvalbumin/Somatostatin).

We thank the reviewer for raising this important question. In our study, we utilized the Dlx1/2b::eGFP reporter to broadly label cortical interneurons; however, this system does not distinguish specific interneuron subtypes. To address this, in the manuscript we used the single-cell RNA sequencing data and immunostainings to provide this information. As expected based on our previous reports, most cortical interneurons present in organoids are represented by calretinin (CALB2), somatostatin (SST) and calbindin (CALB1). These data are now presented in Fig. S3.

Separately, we used available scRNA-seq data from developing human brain and showed that at ~20 PCW, the developing human brain has similar types of cortical interneurons. These data are now included in Fig. S5.

(2) The authors should test more candidates from their bulk RNA-seq data with different fold changes for regulation after hypoxia, to allow the reader to judge at which cut-off the DEGs may be reproducible. This would make this database much more valuable for the field of hypoxia research.

We appreciate the reviewers’ thoughtful suggestion. In addition to the bulk RNA-seq analysis, we did validate several upregulated hypoxia-responsive genes with varying fold changes by qPCR; these include PDK1, PFKP, VEGFA (Fig. S1).

We do agree that in-depth investigation of specific cut-offs would be interesting, however, this could be the focus of a different manuscript.

**Reviewer #3 (Recommendations for the authors):**
Most of the evidence presented is convincing in supporting the conclusions, and I have only minor suggestions for improvement:(1) The bulk RNA-seq was performed in hSOs only, which may not fully capture the phenotypes of migrating or migrated interneurons. It would be valuable, if feasible, to sort migrated cells from hSO-hCO assembloids and specifically examine their molecular mediators.

We thank the reviewer for this suggestion. While it is likely that the cellular environment will have some influence on a subset of the molecular changes, based on all the data from the manuscript and our specific target, the RNA-sequencing on hSOs was sufficient to capture essential changes like ADM upregulation. The in-depth exploration on differential responses of migrated versus non-migrated interneurons to hypoxia could be the focus of a different project.

(2) In Figure 3, it is striking that cell-type heterogeneity dominates over hypoxia vs. control conditions. A joint embedding of hSO and hCO cells could provide further insight into molecular differences between migrated and non-migrated interneurons.

We thank the reviewer for this observation and opportunity to clarify. Since we manually separated the assembloids before the analyses, we processed these samples separately. That is why they separate like this. In the revision, we added data about ADM expression and its receptors’ expression in the hCOs.

(3) It would be helpful to expand the discussion on how closely the migration observed in hSO-hCO assembloids reflects in vivo conditions, and what environmental aspects are absent from this model. This would better frame the interpretation and translational relevance of the findings.

We thank the Reviewer for bringing up this important point. Although the assembloid model offers the unique advantage of allowing the direct investigation of migration patterns of hypoxic interneurons, we fully agree it does not fully recapitulate the in vivo environment. While there are multiple aspects that cannot be recapitulated in vitro at this time (e.g. cellular complexity, vasculature, immune response, etc), we are encouraged by the validation of our main findings in ex vivo developing human brain tissue, which strongly supports the validity of our findings for in vivo conditions.

We expanded our discussion to include more details and the need to validate these findings using in vivo models.

(4) The authors suggest that hypoxia is also associated with delayed interneuron maturation, yet the bulk RNA-seq data primarily reveal stress and hypoxia-related genes. A more detailed discussion of why genes linked to interneuron maturation and function were not strongly affected would clarify this point.

We thank the Reviewer for the opportunity to clarify.

The RNAseq data was performed during the acute stages of hypoxia/reoxygenation and we think a maturation phenotype might be difficult to capture at this point and would require analysis at later in vitro assembloid maturation stages.

Our speculation about a possible maturation defect is based on data from previous studies from developmental biology that showed failure of interneurons to reach their final cortical location within a specified developmental window will impair their integration within the neuronal network, and thus lead to maturation defects and possible elimination by apoptosis.

Since preterm infants suffer from countless hypoxic events over multiple months, we speculate these repetitive events are likely to induce cumulative delays in migration, inability of interneurons to reach their target in time, followed by abnormal integration within the excitatory network, and eventual elimination of some of these interneurons through apoptosis. However, the direct demonstration of this effect following a hypoxic insult would require prolonged in vivo experiments in rodents to follow the migration, network integration and apoptosis of interneurons; to our knowledge this experimental design is not technically feasible at this time, and thus this hypothesis remains speculative and only included in the discussion.

(5) Relatedly, while the focus on interneuron migration is well justified, acknowledging how hypoxia might also impact other aspects of cortical development (e.g., progenitor proliferation, neuronal maturation, or circuit integration) would place the findings in a broader developmental framework and strengthen their relevance.

We appreciate the Reviewer’s suggestion to discuss the role of hypoxia on other interneuron developmental processes during cortical development. In the manuscript, we included text in the discussion about the likely effects of hypoxia on interneuron proliferation, maturation and circuit integration.

(6) Very minor: in Figure S3C and D, it was not stated what the colors mean (grey: control, yellow: hypoxia)

Thank you for pointing out this error; we corrected it in our revision.